



# A quantitative decoupling analysis (QDA v1.0) method for the assessment of meteorological, emission and chemical contributions to fine particulate pollution

Junhua Wang[1,3], Baozhu Ge[*,1,3], Xueshun Chen[1,3], Jie Li[1,3], Keding Lu[2], Yayuan Dong[1,3], Lei Kong[1,3],
Zifa Wang[*,1,3], Yuanhang Zhang[2]

[1]State Key Laboratory of Atmospheric Boundary Layer Physics and Atmospheric Chemistry (LAPC), Institute of Atmospheric Physics (IAP), Chinese Academy of Sciences (CAS), Beijing 100029, China
[2]College of Environmental Sciences and Engineering, Peking University, Beijing, 100871, China
[3]College of Earth and Planetary Sciences, University of Chinese Academy of Sciences, Beijing, 100049, China

*Correspondence to*: Baozhu Ge (gebz@mail.iap.ac.cn) and Zifa Wang (zifawang@mail.iap.ac.cn)

**Abstract.** A comprehensive understanding of the effects of meteorology, emission and chemistry on severe haze is critical in the mitigation of air pollution. However, such understanding is largely hindered by the nonlinearity of atmospheric chemistry systems. Here, we developed a novel quantitative decoupling analysis (QDA) method to quantify the effects of emission, meteorology, chemical reaction, and their nonlinear interactions on the fine particulate matter ($PM_{2.5}$) pollution based on the

accompanying simulations for different atmospheric processes. Via embedding the QDA method into the Weather Research and Forecasting-Nested Air Quality Prediction Modeling System (WRF-NAQPMS) model, we first employed this method into a typical heavy haze episode in Beijing. Different from the previously sensitive simulation method, which usually linked to a certain period, the QDA achieves the fully decomposing analysis of $PM_{2.5}$ concentration during any pollution event into seven different parts, including meteorological contribution (M), emission contribution (E), chemical contribution (C), and

interactions among these drivers (i.e., ME, MC, EC and MCE). The results show that the meteorology contribution varied significantly at different stages of episode, from 0.21 $\mu g \cdot m^{-3} \cdot h^{-1}$ during accumulation period to -11.82 $\mu g \cdot m^{-3} \cdot h^{-1}$ during the removal period, dominating the hourly changes of $PM_{2.5}$ concentrations. The chemical contributions were shown to increase with the level of haze, which become largest (0.37 $\mu g \cdot m^{-3} \cdot h^{-1}$) at the maintenance period, 25% higher than that during the clean period. The contribution of primary emission is relatively stable in all stages due to the use of fixed emission during the

simulation. Besides, the QDA method highlights that there exist nonnegligible coupling effects of meteorology, emission and chemistry on $PM_{2.5}$ concentrations (-1.83 to 2.44 $\mu g \cdot m^{-3} \cdot h^{-1}$), which were commonly ignored in previous studies and the development of heavy-pollution control strategies. These results indicate that the QDA method can not only provide researchers and policy makers with valuable information for understanding of key factors to heavy pollution, but also help the modelers to find out the sources of uncertainties among numerical models.





## 1 Introduction

Atmospheric particulate matter especially fine particulate matter smaller than 2.5 µm (PM$_{2.5}$), can reduce visibility, degrade air quality, boost health expenditures, and increase respiratory diseases and mortality (Xing et al., 2021; Huang et al., 2014; Lelieveld et al., 2015). Over the past two decades, rapid development of industrialization and urbanization has led to severe haze pollution in China (Lu et al., 2019b; Chen et al., 2018; Liu et al., 2017; Hartmann et al., 2014) with the Beijing-

Tianjin-Hebei (BTH) region exhibiting the largest PM$_{2.5}$ concentrations (Lin et al., 2015). The ambient PM$_{2.5}$ concentration is controlled by complex atmospheric processes, including emission, meteorology and chemical reactions (Gelencsér et al., 2007; Jia et al., 2015; Wang et al., 2015; He et al., 2016; Sun et al., 2016). Thus, an effective PM$_{2.5}$ control strategies should be formulated and adopted on the basis of an in-depth understanding of the effects of meteorology, emission, chemistry and their interactions on the formation of PM$_{2.5}$. However, it is difficult to quantify and distinguish the roles of each factor from the

others due to their complex mechanisms and varied conditions from case to case (Li et al., 2011).

There have been some tools developed to analyse the effects of different factors on PM$_{2.5}$ concentrations. The integrated process rate (IPR) considered in the Community Multiscale Air Quality (CMAQ) model can analyse the contribution of physicochemical processes in numerical models, thus providing a comprehensive understanding of the formation of air pollution (Jeffries and Tonnesen, 1994). IPR methods have been applied to study the formation process and mechanism of

ozone and particulate matter in many cities (Liu et al., 2010; Li et al., 2014; Fan et al., 2014; Huang et al., 2016; Chen et al., 2019a; Chen et al., 2019c; Fu et al., 2020). The source apportionment technique (SAT) can quantitatively estimate the contributions of certain air pollutant emissions originating from different regions to the target grid (Lv et al., 2021; Lu et al., 2017; Li et al., 2017b). The above methods, however, cannot accurately reveal the influence of meteorological, emission and chemical effects on PM$_{2.5}$ during heavy-pollution periods. Besides, the scenario analysis approach (SAA) has been employed

to assess the response of PM$_{2.5}$ by changing emissions under certain meteorological fields, For example the studies of Zheng et al. (2015b) found that the heavy pollution occurring in winter 2013 was mainly caused by the stable weather conditions in most parts of Northeast China rather than driven by a sudden increase in anthropogenic emissions. Zhang et al. (2019a) reported that although interannual meteorological changes may notably affect the PM$_{2.5}$ concentration, the corresponding impact on the five-year trend is relatively limited. However, due to the nonrepeatability of individual pollution cases, sensitivity experiments

considering meteorological conditions or emission changes cannot fully reproduce the individual cases.

In addition to chemical transport model (CTM) simulations, certain other methods have also been adopted. For example, the PLMA (parameter linking air quality to meteorological conditions) index has been used to determine the contribution of meteorology and emission to air pollution (Zhang et al., 2015; Zhang et al., 2019b; Yang et al., 2016). Studies employing principal component analysis or those targeting the correlation between PM$_{2.5}$ and meteorological elements suggested that a

low wind speed and high humidity facilitate haze formation (Wang et al., 2013; Pang et al., 2009; Shu et al., 2017; Zhai et al., 2019). Considering that a single meteorological element does not fully explain the relationship between meteorology and PM$_{2.5}$,



an artificial neural network (ANN) model has been used to combine multiscale meteorological conditions, and the meteorological influence has thus been quantified by the explained variance (He et al., 2017).

None of the above methods quantitatively analyses individual pollution cases nor provides a direct grasp of environmental management to decision makers. Although the basic relationship between $PM_{2.5}$ and different influencing factors has been revealed, the quantitative influence of these factors on pollution episodes remains unclear. In view of these discrepancies, in this paper, we developed a novel quantitative decoupling analysis (QDA) method and for the first time assess the effects of emissions, meteorology, chemical reactions as well as their interactions on the $PM_{2.5}$ concentration in a pollution case over BTH region. Briefly, the QDA method tracks the change in $PM_{2.5}$ concentration in response to changes in emissions, meteorological conditions and chemical reactions in high-pollution cases. Based on QDA, the roles of the above three factors and their mutual coupling influence are quantified. Thus, this approach provides a useful tool to determine the air pollution control and emergency responses in pollution cases in typical urban areas. It can also help identify the key process and improve its representation in atmospheric models, for example, the physicochemical structure in boundary layer and formation mechanism of secondary air pollution (Chen et al., 2019a; Kang et al., 2019; Xing et al., 2017; Goncalves et al., 2009).

## 2 Description of the quantitative decoupling analysis (QDA) method

In this section, we provide a detailed description of the QDA method proposed in this study, including the general design of QDA method and its implementation, the chemical transport model used in this study and the experiment set up for the application of the QDA method into the case study over BTH region.

### 2.1 Design of the QDA method

Emissions, meteorology and chemistry are the three main drivers of $PM_{2.5}$ variation in the atmosphere. They also interact with each other due to the nonlinear characteristics of atmospheric systems, which should be considered in the quantification of their contribution to $PM_{2.5}$ changes. Thus, in the QDA method, we considered that the $PM_{2.5}$ concentration is a function of the above three driving factors, and explicitly considers the interactions among these three drivers by representing the $PM_{2.5}$ concentration as $F(\vec{x})$, and the variation in $PM_{2.5}$ can be expressed as Eq. (1):

$$\Delta PM_{2.5} = F(\vec{x} + \overrightarrow{\Delta x}) - F(\vec{x}); \ \vec{x} = (x_1, x_2, x_3) \tag{1}$$

where $\vec{x}$ is a given influencing factor of $PM_{2.5}$ and $x_1, x_2,$ and $x_3$ denote the meteorological, emission and chemical factors, respectively. $F(\vec{x} + \overrightarrow{\Delta x})$ is the $PM_{2.5}$ concentration upon $x_i$ (i=1,2,3) variation. According to the Taylor equation we can obtain Eq. (2):





$$F(\vec{x} + \overrightarrow{\Delta x}) - F(\vec{x}) = \sum_{i=1}^{3} \frac{\partial F}{\partial x_i} \Delta x_i + \frac{1}{2!}\left(\sum_{i=1}^{3} \frac{\partial^2 F}{\partial x_i^2} \Delta x_i{}^2 + \frac{\partial^2 F}{\partial x_1 \partial x_2} \Delta x_1 \Delta x_2 + \frac{\partial^2 F}{\partial x_2 \partial x_3} \Delta x_2 \Delta x_3 + \frac{\partial^2 F}{\partial x_1 \partial x_3} \Delta x_1 \Delta x_3\right) +$$

$$\frac{1}{3!}\left(\sum_{i=1}^{3} \frac{\partial^3 F}{\partial x_i^3} \Delta x_i{}^3 + \sum_{a=1}^{2} \frac{\partial^3 F}{\partial x_1{}^a \partial x_2{}^{3-a}} \Delta x_1{}^a \Delta x_2{}^{3-a} + \sum_{a=1}^{2} \frac{\partial^3 F}{\partial x_2{}^a \partial x_3{}^{3-a}} \Delta x_2{}^a \Delta x_3{}^{3-a} + \sum_{a=1}^{2} \frac{\partial^3 F}{\partial x_1{}^a \partial x_3{}^{3-a}} \Delta x_1{}^a \Delta x_3{}^{3-a} +$$

$$\frac{\partial^3 F}{\partial x_1 \partial x_2 \partial x_3} \Delta x_1 \Delta x_2 \Delta x_3\right) + \cdots + o^n \ (\text{n>3}) \tag{2}$$

The terms only containing a single partial derivative to $x_1$ (including any higher-order derivatives) are denoted as M, where M indicates the meteorological contribution without consideration of any coupling effects. In the same manner, the partial derivative terms of $x_2$ are denoted as E, representing the emission contribution without consideration of any coupling

effects. The partial derivative terms of $x_3$ are denoted as C, representing the chemical contribution without coupling effect consideration. The other terms represent the interaction among the different drivers: the partial derivatives of $x_1$ and $x_2$ reflect the effect of emissions on meteorology (ME), the partial derivatives of $x_2$ and $x_3$ reflect the effects of emissions on chemistry (CE), the partial derivatives of $x_1$ and $x_3$ reflect the interactions between meteorology and chemistry (MC) and MCE denotes the interactions among the above three drivers. Therefore, Eq. (2) can be rewritten as Eq. (3):

$$F(\vec{x} + \overrightarrow{\Delta x}) - F(\vec{x}) = M + E + C + ME + CE + MC + MCE \tag{3}$$

The relationship among the 7 terms of Eq. (3) can be expressed with the graph theory, as illustrated in the Fig. 1, in which the contributions of emissions, meteorology and chemistry are denoted by three circles and the interactions among the different processes are denoted by overlapping areas. The arrows indicate the directions of the coupling effects between any two drivers. Intuitively, the effect of emissions on meteorology or chemical processes is unidirectional, while the effect between

meteorology and chemistry is bidirectional.

According to these definitions, each term in Eq. (3) can be determined via scenario simulations. For example, the magnitude of M can be determined via scenario simulations without considering emission and chemical processes. Moreover, the magnitude of M+E+ME can be determined with the scenario simulation method without chemical processes. Based on this method, we designed six accompanying simulation scenarios and embedded them into the baseline simulation scenario (Table

1) to calculate each term of Eq. (3). The baseline scenario (M1) represents the actual situation, which is employed to calculate the change in the PM$_{2.5}$ concentration at hourly intervals. The six accompanying simulation scenarios represent simulations without one or two drivers that are performed to decompose the PM$_{2.5}$ variation. Here, the accompanying means that the simulation scenarios are simultaneously run with the base simulation scenario at each time step. Thus, in contrast to the traditional scenario simulation approach wherein each simulation scenario is independently run with the baseline simulation

scenario, the accompanying simulation scenarios are encoded in the baseline simulation scenario. With the use of the results of the baseline and different accompanying scenario simulations, the magnitudes of M, C, E, ME, CE, MC, and MCE at each time step can be easily calculated.

The above QDA method can also be combined with the IPR method to resolve more detailed information, such as the contributions of advection, diffusion, dry and wet deposition processes in M or the contributions of the gas-phase chemistry,

thermodynamic equilibrium processes and reactions involving secondary organic aerosols (SOAs) in C. Detailed descriptions





about these processes are available in Table 2. In addition, this technique enables us to derive the proportion of various physicochemical processes in the different coupling interactions, which can be considered to explore the causes of air pollution generation, maintenance and dissipation and determine the different characteristics of clean and polluted periods.

## 2.2 Chemical transport model

It should be noted that the QDA method designed in this study is a universal tool that can be embedded into any numerical model, which is another advantage of this method. In this study, we embedded the QDA method into the Nested Air Quality Prediction Modeling System (NAQPMS) model. NAQPMS is a three-dimensional regional Eulerian chemical transport model developed by the Institute of Atmospheric Physics, Chinese Academy of Sciences, which has been widely used in scientific research and air quality prediction practice (Wang et al., 2014) due to its good performance in simulating the emission,

meteorological and chemical processes in the atmosphere. Within the model, the gas-phase chemistry was simulated by carbon bonding mechanism Z (CBM-Z) developed by Zaveri and Peters (1999) which includes 134 reactions and 71 species. For inorganic aerosols, the thermodynamic equilibrium module ISORRPIA v1.7 (Nenes et al., 1998) was used to simulate the ammonia-nitrite-sulfate-chloride-sodium-water system. Six secondary organic aerosols (SOA) were processed by a two-product module in NAQPMS (Odum et al., 1997). The aqueous-phase chemistry and wet deposition are modelled using the

Regional Acid Deposition Model (RADM) mechanism in the Community Multi-scale Air Quality (CMAQ) version 4.6. Dry deposition of gases and aerosols is simulated based on the scheme of Wesely (1989) and the advection is simulated with an accurate mass-conservative algorithm from Walcek and Aleksic (1998). More technical details on NAQPMS could be found in Li et al. (2012).

## 2.3 Experiment setup

To illustrate the feasibility of the QDA method and quantitatively resolve the magnitude of the emission, meteorological and chemical contributions to the $PM_{2.5}$ variation during heavy-pollution episode, we applied it into a week-long heavy-haze episode in Beijing from 17 to 28 February. Figure 2 shows the modelling domain of this case, which covers the most parts of East Asia with a horizontal resolution of 45km. Vertically, the NAQPMS uses nonequally distributed 20 terrain-following layers from the surface to 20km. The anthropogenic emission inventories used in the simulation were obtained from the

Chinese multi-resolution emission inventory (MEIC) developed by Tsinghua University (http://www.meicmodel.org). Biogenic emissions were obtained from the National Center for Atmospheric Research (NCAR), derived from the model of natural gas and aerosol emissions (MEGAN v2.0)(Guenther et al., 2006). The clean initial condition was used in the simulation with a 10-day free run of NAQPMS as a spin-up time. Top and boundary conditions of the outermost region were extracted from the global chemical transport model MOZART-v2.4 (The Model for Ozone and Related Chemical Tracers version 2.5)

with 3-hour time resolution(Brasseur et al., 1998). Hourly meteorological fields were generated by Weather Research and Forecasting Model version 3.7 (WRFv3.7) (http://www.wrf-model.org/) driven by the National Centre for Environmental Prediction (NCEP) Final Analysis (FNL) data.





## 2.4 Observations

The observational data used in this study include surface observations of PM$_{2.5}$, fine particulate matter smaller than 10 µm (PM$_{10}$), NO$_2$, O$_3$, SO$_2$ and CO obtained from the China National Environmental Monitoring Center, surface observations of the wind speed, wind direction, temperature, relative humidity and station pressure, and vertical observations of the wind speed, wind direction, temperature, and relative humidity retrieved from the China Meteorological Administration. Spatial distributions of the meteorological and air quality observation sites are shown in Fig. 2. To compare with the PM$_{2.5}$ observations, The simulated PM$_{2.5}$ concentrations comprised of primary PM$_{2.5}$, including black carbon, primary organic aerosol, and other

directly emitted PM$_{2.5}$, and secondary PM$_{2.5}$, including sulfate, nitrate, ammonium and SOA produced by chemical reactions.

## 3 Results

### 3.1 Observed pollution during the heavy-haze episode

From 19–27 February 2014, a serious pollution event occurred in the Beijing area, with the average PM$_{2.5}$ concentration observed at the different sites in Beijing reaching 222.4 µg m$^{-3}$ (Fig. 3), nearly three times higher than the national secondary

standard level (75 µg m$^{-3}$). As shown in Fig. S1, this pollution episode also affected a wide area of the BTH region, with severe haze mostly located in the southern part of the BTH region before 23 February and gradually extending northward to encompass wider areas. The SO$_2$, NO$_2$ and O$_3$ concentrations did not exhibit notable exceedances as the PM$_{2.5}$ did, indicating that this case is a typical particulate-led pollution event.

To investigate the variation in the contributions of emissions, meteorology and chemistry at the different stages of this

haze event, we divided the whole episode into four stages based on the temporal characteristics of the PM$_{2.5}$ concentration in Beijing (Fig. 3): (1) the pre-contamination stage (February 17/08:00–19/14:00 LST) when the PM$_{2.5}$ concentration was low and its variation was limited, representing a relatively clean period; (2) the accumulation stage (February 19/15:00–23/08:00 LST) when the PM$_{2.5}$ concentration increased the most rapidly; (3) the pollution maintenance stage (February 23/09:00–26/18:00 LST) when the PM$_{2.5}$ concentration remained high with small fluctuations; and (4) the pollution removal stage

(February 26/19:00–27/08:00 LST) when the PM$_{2.5}$ concentration rapidly dropped.

### 3.2 Evaluation of the meteorology and chemistry simulations

To assess the accuracy of the model, simulated meteorological parameters and air pollutant concentrations were compared with observed values. We use several evaluation indicators to quantitatively assess model performance, including Simulated average (MM), Observed average (OM), correlation coefficient (R), mean fractional bias (MFB), mean deviation (MB),

standard mean deviation (NMB), standard mean error (NME), root mean square error (RMSE), and index of agreement (IOA), which are defined in Table S1. The verification results of meteorological elements are shown in Table S2, and the correlation coefficient (R) of temperature (Temp), relative humidity (RH) and pressure are all above 0.85. The correlation between wind





speed (WS) and observation data (R=0.47) is better than that of wind direction (WD:R=0.24). Although the error of wind simulation is greater than that of other meteorological elements, NME and NMB are less than 1, which indicates that simulation

and observation matches well on the whole, and the error may have little influence on the performance of aerosol simulation.

The simulations based on the NAQPMS model generally reproduced the magnitude and temporal variation in the $PM_{2.5}$ concentration in the Beijing area well, with a correlation coefficient (R) of approximately 0.83 and a mean bias error (MBE) smaller than 1 µg m$^{-3}$. The model simulation results exhibited relatively larger underestimations of the $PM_{2.5}$ concentration from 20–23 February, which may be attributed to the overestimation of the simulated wind speed by the WRF model during

this period (Fig. 4~5). In regard to the two important precursors of $PM_{2.5}$, the simulated $NO_2$ and $SO_2$ concentrations also agreed well with the observations, with MBE values of approximately 7.1 and 5.3 µg m$^{-3}$, respectively, and R values of approximately 0.75 and 0.74, respectively. In general, the simulated $PM_{2.5}$, $SO_2$ and $NO_2$ concentrations all satisfactorily satisfied the mean fractional bias (MFB) and mean fractional error (MFE) prime performance standards (MFB:30%;MFE:50%) (Boylan and Russell, 2006). The simulated sulfate, nitrate and ammonium concentrations were also evaluated against the

observations to evaluate the chemical processes in the NAQPMS model (Fig. 6). The model reproduced the variation in secondary inorganic aerosols (SIAs) well during this episode (R>0.79), although the model underestimated the sulfate concentration, possibly due to missing reaction pathways of sulfuric acid in the model, such as heterogeneous chemistry (Zheng et al., 2015a; Cheng et al., 2016). Underestimation of the sulfate concentration is a common problem in current CTMs (Chen et al., 2019b), which is beyond the scope of this study. However, this could lead to uncertainty in the estimation of the chemical

contribution to the $PM_{2.5}$ concentration in this study. In summary, the model can suitably reproduce the pollution evolution process from the cleaning period to the accumulation, maintenance and removal periods, which lays a good foundation for the subsequent analysis of the physical and chemical processes.

## 3.3 Illustration of the QDA results

### 3.3.1 Temporal variation in the QDA results

Figure 7 shows the time series of the calculated contributions of emissions, meteorology, chemistry and their interactions to the hourly variation in the $PM_{2.5}$ concentration using the QDA method. We can clearly observe that the sum of all resolved contributions is exactly equal to the hourly change in the $PM_{2.5}$ concentration, indicating that the QDA method can fully resolve the variation in the $PM_{2.5}$ concentration. According to the QDA results, the fluctuation in $PM_{2.5}$ caused by the meteorological factor is the most significant, with calculated meteorological contributions (M) ranging from -42 to 8 µg·m$^{-3}$·h$^{-1}$. At the hourly

scale, emissions are the second largest contributor to the $PM_{2.5}$ variation. Since we did not consider the temporal variation in the emission inventory in this case, the calculated emission contribution (E) remains constant (0.9 µg·m$^{-3}$·h$^{-1}$) throughout the episode. The chemical contribution exhibits remarkable diurnal variation, which is notably larger during the daytime than that during the nighttime. This occurs because the atmospheric oxidation capacity during the daytime is higher than that during the





nighttime, which is more conducive to secondary $PM_{2.5}$ formation (Huang et al., 2021; Chen et al., 2020a; Lu et al., 2019a),
and similar conclusions have been reported in other model studies (Chen et al., 2019a; Li et al., 2014).

Figure 8 shows the QDA results at the different stages of the episode. The meteorological processes exhibited a notable negative contribution to $PM_{2.5}$ at the first stage, which was enough to remove the newly emitted or formed $PM_{2.5}$ from emissions and chemical reactions. Thus, the $PM_{2.5}$ concentration was relatively low at the first stage. However, M shifted to a positive value at stage 2, and almost no removal processes occurred during this period, leading to rapid accumulation of the $PM_{2.5}$
concentration. M became negative and played a clearing role at stage 3, but it nearly offset the simultaneous increase in emissions and chemistry. Hence, the $PM_{2.5}$ concentration remained at a relatively steady level. At stage 4, the meteorological removal effects were much larger than those at the previous stages due to the strong northwest nonpolluted wind, leading to a rapid decline in the $PM_{2.5}$ concentration. According to the process analysis results, horizontal advection was the main process of pollution removal at stages 1 and 4, and a high horizontal wind speed during the cleaning period facilitated $PM_{2.5}$ reduction
(Chen et al., 2020b). Vertical advection was the main process of pollution accumulation at stages 1, 2 and 4, indicating the transportation of fine particles from upper to lower altitudes. High pollutant levels may originate from elsewhere via long-distance transport (Du et al., 2020), leading to boundary layer turbulence weakening, which facilitates local pollution accumulation (Huang et al., 2020). The vertical diffusion contribution in M was less than zero at all stages, indicating that the considered pollutants continuously diffused from lower to upper altitudes and that this feature did not change at the different
stages. In addition, the hourly contribution of deposition was small and played a negligible role in the observed variation at the different stages of pollution. The chemical processes yielded positive contributions at the first three stages (0.29-0.37 $\mu g \cdot m^{-3} \cdot h^{-1}$) due to the generation of SIAs and SOAs. At stage 4, the chemical contribution became negative (-0.18 $\mu g \cdot m^{-3} \cdot h^{-1}$), as the environmental conditions at this stage were suitable for nitrate decomposition (Chen et al., 2020b).

In summary, the QDA results suggest that the main sources of $PM_{2.5}$ are primary emission processes (E) and chemical
processes (C). Meteorological processes (M) act as the main removal pathway of $PM_{2.5}$ in the atmosphere, and the different atmospheric motion states and pollutant concentrations in the atmosphere can cause positive or negative M contributions during the different periods, which exerts the largest impact on the hourly variation in the $PM_{2.5}$ concentration. When M becomes positive, no $PM_{2.5}$ removal process occurs, and combined with emissions and chemical processes, leading to a rapid increase in $PM_{2.5}$.

**3.3.2 Coupling effect at the different stage**

There are also nonnegligible coupling effects at each stage according to the QDA method despite their contributions being smaller than those of meteorology, emission and chemistry (Fig. 7~8). For example, when the emission process increases the concentration of pollutants in air, the amount of pollutants carried by air masses also increases, which is conducive to pollutant transport, reflecting the coupled effect of EM (-0.14 ~ -0.06 $\mu g \cdot m^{-3} \cdot h^{-1}$). The emission process also increases the concentration
of precursors in the atmosphere. Hence, more secondary aerosol particles are generated through chemical reactions, as reflected by CE>0 at each stage. Based on the above process analysis results, this coupling effect is largely associated with the generation





of SIAs via chemical reaction and equilibrium partition. The interaction between chemistry and meteorology (MC) consists of two parts: the first part is the influence of meteorology on chemistry, in which meteorological processes can increase chemical production by transporting more precursors or decrease chemical production by reducing the concentration of local precursors,

while the second part involves the influence of chemistry on meteorology, since chemical processes can lead to an increase in the concentration of secondary pollutants in the atmosphere. This may lead to an increase in pollutants carried by air masses in the corresponding region. The coupling term MCE includes all meteorological, emission and chemical process interactions, which are complex but yield very small contributions.

Based on these results, we investigated for the first time the influences of the coupling effects on the $PM_{2.5}$ concentration

by summing all the contributions of the interaction between the different physical and chemical processes (COUP=EM+CE+MC+MCE). The hourly value of COUP ranged from -1.83 to 2.44 $\mu g \cdot m^{-3} \cdot h^{-1}$ during this episode, with an average value of approximately 0.30 $\mu g\ m^{-3}\ h^{-1}$. The coupling effect was shown to increase continuously from the relatively clean to polluted periods. From stages 1 to 3, the hourly average value of COUP increased from 0.13 to 0.5 $\mu g \cdot m^{-3} \cdot h^{-1}$, but its proportion in the hourly variation in $PM_{2.5}$ decreased continuously (from 37% to -3%). This indicates that although the coupling

effect increased with the $PM_{2.5}$ concentration, the influence of the other effects increased faster than the coupling effect did.

During the whole episode, CE exhibited the largest coupling effect (0.27 $\mu g \cdot m^{-3} \cdot h^{-1}$ on average) and increased with $PM_{2.5}$ concentrations, indicating that the coupling between emissions and chemistry plays an important role during periods of heavy haze. According to the vertical distribution of CE at stage 2 (Fig. 9), the contribution of CE decreased from the surface to the upper levels due to the vertical characteristics of the air temperature and emissions. The contribution of MC revealed the largest

variation with a fluctuation range up to 4.24 $\mu g \cdot m^{-3} \cdot h^{-1}$ because both the meteorology and chemistry are greatly influenced by diurnal variation. As shown in Fig. 9, MC also indicated that meteorological processes could decrease the chemical formation process at the surface and increase chemical formation in the upper layers (L3~L8), which could be related to the phenomenon whereby meteorological action transports $PM_{2.5}$ (M) and precursors from the bottom layer to the upper layer. EM suggests that local emissions could enhance vertical $PM_{2.5}$ diffusion from the surface layer to the upper layer. Primary emitted $PM_{2.5}$ particles

mainly occurred in the near-surface layer, where the vertical wind speed was so low that vertical advection was very limited. Thus, $PM_{2.5}$ emitted by pollution sources could only diffuse to the upper layers. MCE represents the interaction between emissions, meteorology, and chemical processes, which contributed little to the $PM_{2.5}$ variation in this case.

In previous studies, examination of coupling effects has usually been ignored in the analysis of heavy-haze episode. The QDA results demonstrated that ignoring these coupling effects may cause deviation when studying the contribution of

meteorological, emission or chemical changes to $PM_{2.5}$. For example, when discussing the effect of emission change on $PM_{2.5}$, if the variations in CE, EM and MCE were ignored, an uncertainty ranging from -0.86~1.86 $\mu g \cdot m^{-3}$ could occur in the hourly results, especially during the most polluted period, and this uncertainty may accumulate with time. This suggests that quantitative analysis of the coupling effect is necessary to the evaluation of the pollution control effect.





### 3.4 Evaluation of the QDA results at different pollution stages

The emission contribution (E) calculated with the QDA method is just directly determined by the emission inventory used in the simulations. Thus, we mainly evaluated the calculated contributions of meteorological and chemical processes in this study. However, there were no observation data directly linked to the emission or chemical contribution that could be used to verify the QDA method directly. Hence, the method was evaluated with indirect results. Since the chemical contribution to $PM_{2.5}$ is mainly related to the formation of secondary aerosols, the conversion rates of nitrate (NOR) and sulfate (SOR), as

defined in Eqs. (5–6), are calculated to evaluate the temporal variation in the chemical contribution obtained with the QDA method. Daily $PM_{2.5}$ mass composition data measured by the Beijing Ecological Environment Monitoring Center are used to calculate NOR and SOR values at the different stages of this episode:

$$NOR = \frac{NO_3^-}{NO_3^- + NO_2} \tag{5}$$

$$SOR = \frac{SO_4^{2-}}{SO_4^{2-} + SO_2} \tag{6}$$

It should be noted that according to the definitions of NOR and SOR, these two indicators are easily affected by regional transportation, as exterior precursors or SIAs may affect the concentration of air pollutants measured locally and reduce the representativeness of NOR and SOR in local chemical transformation. In this case, NOR and SOR increased by 0.09 and 0.02, respectively, from stages 2 to 1. NOR and SOR both reached their maximum values of 0.54 and 0.38 at stage 3. At stage 4, NOR and SOR experienced a significant decrease. Other haze cases have also revealed that SOR and NOR greatly increase

with $PM_{2.5}$ concentration (Song et al., 2019; Xu et al., 2017; Yan et al., 2015a). The proportion of secondary particulate matter often increases with worsening haze (Xu et al., 2019; Li et al., 2017a). Process analysis has also shown that the chemical reaction of $PM_{2.5}$ in the WRF-Chem model is stronger during the day than that at night, which is consistent with the QDA results (Chen et al., 2019a). This evidence combined with the QDA analysis results explains the important role of chemical reactions in severe smog (Huang et al., 2019).

The contributions of meteorological processes were quantitively evaluated via the analysis of weather conditions. Figure S2 in supporting information clearly shows that at stage 1, Beijing and its surrounding areas were influenced by a high-pressure system in northeastern Inner Mongolia and a low-pressure system in the southwest with a high wind speed, which promoted $PM_{2.5}$ advection across the Beijing area. With the low-pressure system in Inner Mongolia slowly moving eastward and finally disappearing under the influence of westerly winds, Beijing was increasingly controlled by a uniform pressure field and

affected by weak south winds, which facilitated the transportation of air pollution from the southern BTH region to Beijing. The small-scale high-pressure center to the north of Beijing also blocked the airflow originating from the south, leading to the accumulation of air pollutants in Beijing, which is consistent with the positive meteorological contribution (M) at stage 2. Although the potential source contribution function (PSCF) index can only reflect the potential contribution of the inflow trajectory, Baoding, Shijiazhuang and Cangzhou in Hebei in southern Beijing were the main sources of $PM_{2.5}$ transmission

(Yan et al., 2015b). Research revealed the transportation process in this case under the influence of weak southerly winds from





February 19-20, and the PLAM index indicated a positive correlation between PM$_{2.5}$ and atmospheric stability (Zhong et al., 2018b). An inversion layer occurred due to the radiative cooling effect of the transported particles, which further aggravated aerosol accumulation (Zhong et al., 2018a) (Fig. 10). In other smog events, the key role of transmission in the formation of high concentrations of PM$_{2.5}$ has also been found (Sun et al., 2016; Huang et al., 2020; Zhang et al., 2019b).

At stage 3, the northern high-pressure system was compressed by the northwest low-pressure air system and moved to the southeast sea area. The isobaric lines in Beijing became increasingly dense, and the wind speed increased, which was beneficial to the diffusion of pollutants (M<0). However, due to the positive contribution of emissions and chemistry, the air quality did not improve. At stage 4, the northeast low-pressure system continued to develop and intensified, confronting the Mongolian high-pressure system, resulting in a strong northwest airflow in North China, which transported air pollutants to

the southeast sea area and greatly improved the air quality in Beijing. Therefore, the hourly contribution of M at this stage was the largest, reflecting a strong cleaning effect. This is also consistent with the analysis of this pollution case in other studies (Zhong et al., 2018b; Zhong et al., 2018a).

## 4 Conclusions and discussions

In this study, a new QDA method targeting PM$_{2.5}$ is proposed and applied to the analysis of a typical heavy-pollution case

in Beijing. By quantitatively decomposing the meteorology, chemical reactions, emission and their coupling interactions in the hourly change in the PM$_{2.5}$ concentration, the formation process of heavy haze can be analyzed from a new perspective.

Through the application of this method into a typical haze episode in Beijing, we found that the meteorological contribution (M) during the accumulation stage (stage 2) was 0.21 μg m$^{-3}$ h$^{-1}$, indicating that M favors the accumulation of PM$_{2.5}$. While M in the removal period (stage 4) plays a strong role in clearance of PM$_{2.5}$. This means meteorological activities

play different roles in different periods, M mainly acts as a sink (negative value) for PM$_{2.5}$ most of the time. When M becomes positive, PM$_{2.5}$ loses its prime sink and is tend to grow rapidly under the superimposed influence of emissions and chemical processes. The contributions of C and CE play a significant role in stage 3, indicating that chemical reactions are more important in the polluted period than in the cleaner period. The contribution of E of PM$_{2.5}$ is independent of other chemical and meteorological factors, because this study did not consider the time variability of the PM$_{2.5}$ primary emission. More

realistic conclusions would be expected in future studies if the temporal variation of emission sources were available.

The method proposed in this study can be applied to different cases with heavy haze characteristics in various cities, which can contribute to the formulation of more effective treatment measures. In addition, the consideration of coupling effects provides a useful way to handle the nonlinear characteristics of the atmosphere, thus fills the gap in traditional methods in terms of nonlinear uncertainty. The QDA method can not only be applied to any three-dimensional atmospheric chemistry

model but can also be employed to study any atmospheric pollutant, including PM$_{2.5}$, which yields a strong general applicability and practical application prospects. This technique provides not only new reference ideas for the governance of air pollution



but also an important tool for the further study of the formation processes of heavy particulate pollution and the influence of different physicochemical mechanisms.

## Code and data availability

The observation data used in this paper and the source codes of QDA are available online via ZENODO (http://doi.org/10.5281/zenodo.5292895). Please contact Junhua Wang (wangjunhua@mail.iap.ac.cn) to obtain the model data for QDA method used in NAQPMS.

## Acknowledgements

This work is funded by the National Natural Science Foundation of China (Grant No. 41877313). We thank the anonymous reviewers for their constructive suggestions that helped improve the manuscript.

## Author contributions

Junhua Wang prepared the original data, designed and conducted the simulation and carried out the QDA method. Baozhu Ge and Xueshun Chen revised the paper and provided scientific guidance for the article design. Yayuan Dong gave advice to the content of the article. Lei Kong provided help for article frame and modified the model code. Yuanhang Zhang, Zifa Wang, KeDing Lu and Jie Li provided valuable suggestions for this article. Junhua Wang wrote the paper and all listed authors have read and approved the final manuscript.

## Competing interests

The authors declare that they have no conflicts of interest.

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



**Figures & tables:**

**Table 1. Grouping mode simulation settings and variable descriptions**

|  | Simulation name | Processes included in the simulations | Target acquisition |
| --- | --- | --- | --- |
| Base simulation | M1 | Base simulation with all physicochemical processes | To quantify the change in the $PM_{2.5}$ concentration at each time step and MCE |
| Accompanying simulations | M2 | Only meteorological process | M |
|  | M3 | Only emission processes | E |
|  | M4 | Only chemical process | C |
|  | M5 | Meteorological and chemical processes | M+C+MC |
|  | M6 | Emission and chemical processes | C+E+CE |
|  | M7 | Emission and meteorological processes | M+E+ME |









**Table 2. Descriptions of different processes considered in the QDA method**

| Abbreviation | Descriptions | Category |
|:---:|:---:|:---:|
| emit | emission | E |
| advhor | horizontal advection | M |
| advvert | vertical advection | M |
| difhor | horizontal diffusion | M |
| difvert | vertical diffusion | M |
| wetdep | wet deposition | M |
| drydep | dry deposition | M |
| gaschem | gas chemistry | C |
| ISORR | inorganic aerosol chemistry | C |
| SOA | secondary aerosol chemistry | C |

**Table 3 Hourly QDA results in different stages (unit: $\mu g\ m^{-3}\ h^{-1}$)**

|  | Stage 1 | Stage 2 | Stage 3 | Stage 4 |
|:---:|:---:|:---:|:---:|:---:|
| **M** | -2.92 | 0.21 | -1.68 | -11.82 |
| **E** | 0.90 | 0.90 | 0.90 | 0.90 |
| **C** | 0.29 | 0.29 | 0.37 | -0.18 |
| **EM** | -0.07 | -0.06 | -0.07 | -0.14 |
| **CE** | 0.14 | 0.15 | 0.43 | 0.50 |
| **MC** | 0.05 | 0.13 | 0.10 | 0.0 |
| **MCE** | 0.02 | 0.02 | 0.05 | -0.13 |





**Table 4. The results of the different-stage process analysis (unit: μg m$^{-3}$ h$^{-1}$)**

|  | emit | advhor | advvert | difhor | difvert | gaschem | drydep | ISORR | wetdep | SOA |
|---|---|---|---|---|---|---|---|---|---|---|
| **Stage 1** | 0.90 | -3.71 | 1.06 | 0.00 | -0.31 | 0.00 | -0.02 | 0.47 | 0.00 | 0.02 |
| **Stage 2** | 0.90 | 0.15 | 0.39 | -0.01 | -0.38 | 0.00 | -0.03 | 0.60 | 0.00 | 0.02 |
| **Stage 3** | 0.90 | 0.13 | -1.31 | -0.01 | -0.55 | 0.00 | -0.03 | 0.93 | 0.00 | 0.04 |
| **Stage 4** | 0.90 | -12.40 | 0.63 | 0.00 | -0.20 | 0.00 | -0.03 | 0.23 | 0.00 | 0.00 |

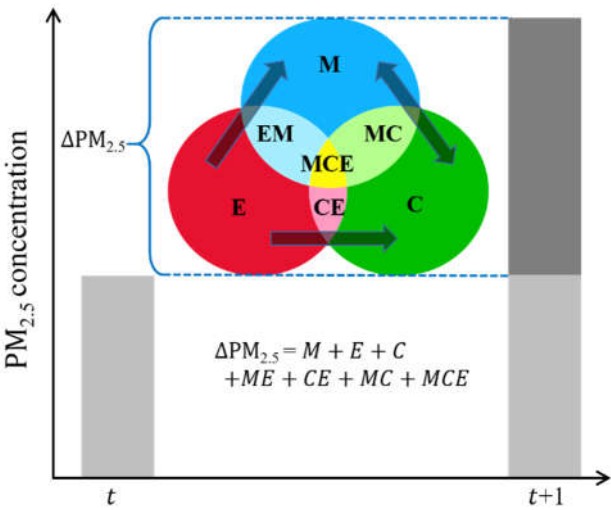


**Figure 1. Graph theory of the QDA method (the total area of colorful graphics represents the change of PM$_{2.5}$ concentration between t and t+1, which can be disassembled into 7 parts, see the Table 2 for abbreviation informations).**

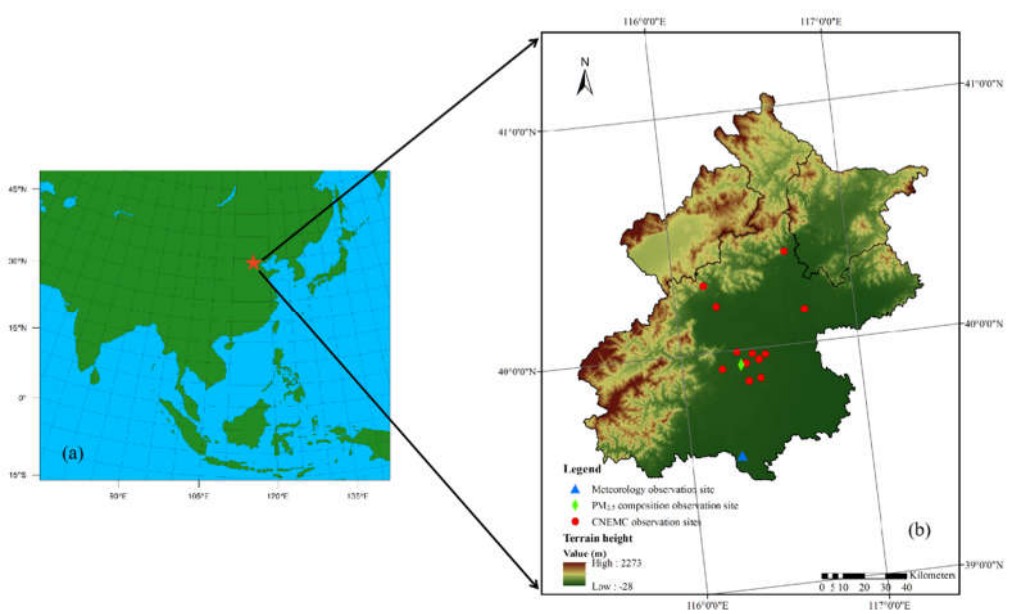

**Figure 2. (a) Model domain and (b) stations for the evaluation used in this study.**

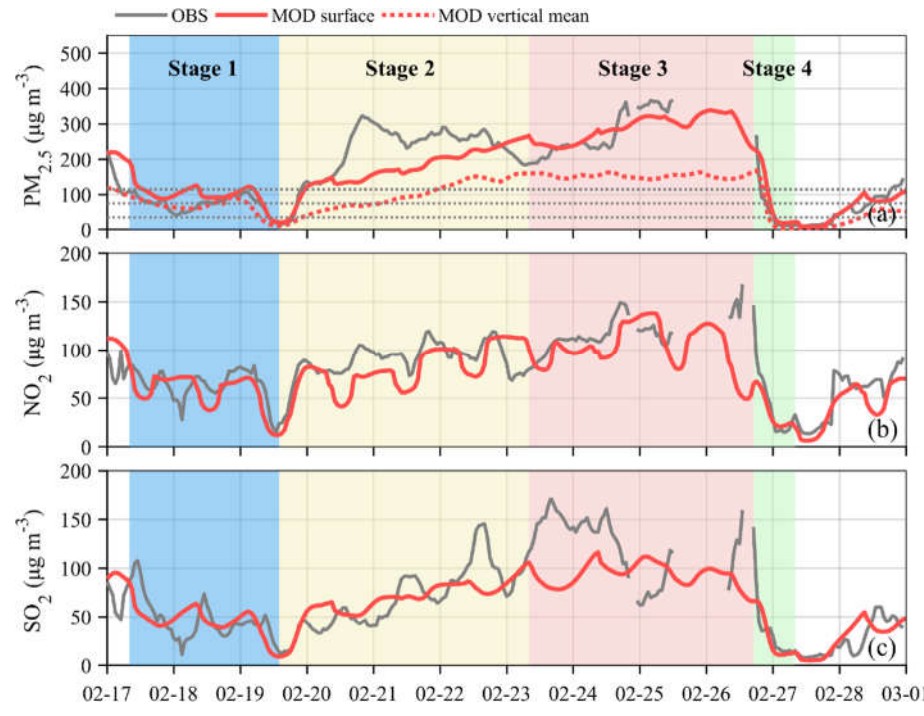

**Figure 3. Observations and simulation results for PM$_{2.5}$, NO$_2$ and SO$_2$ in Beijing (all simulation and observation results are averaged over the Beijing area. The three gray dotted lines indicate 35, 75 and 115 μg m$^{-3}$).**


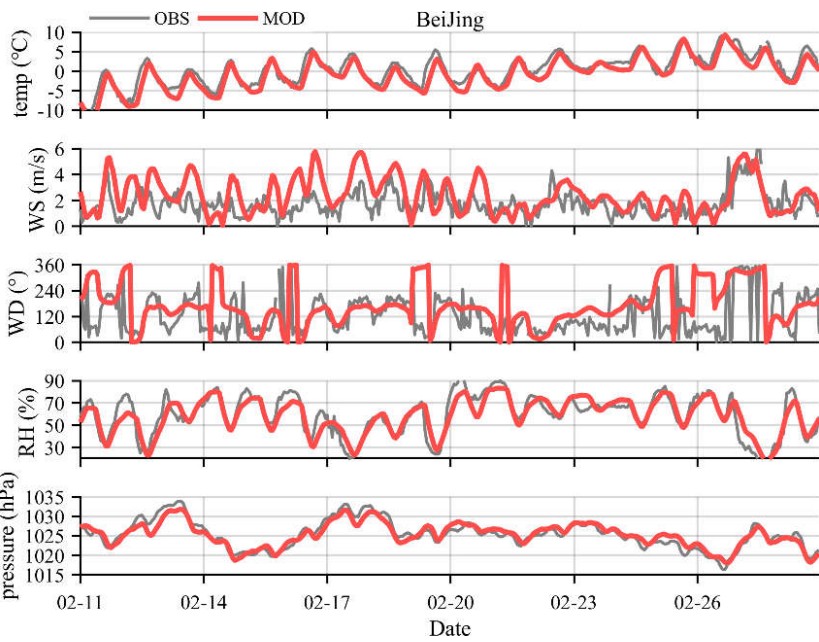

**Figure 4. Comparisons of observed (grey lines) and simulated (red lines) values of different meteorology elements in Beijing from 11th Feb to 28th Feb 2014.**

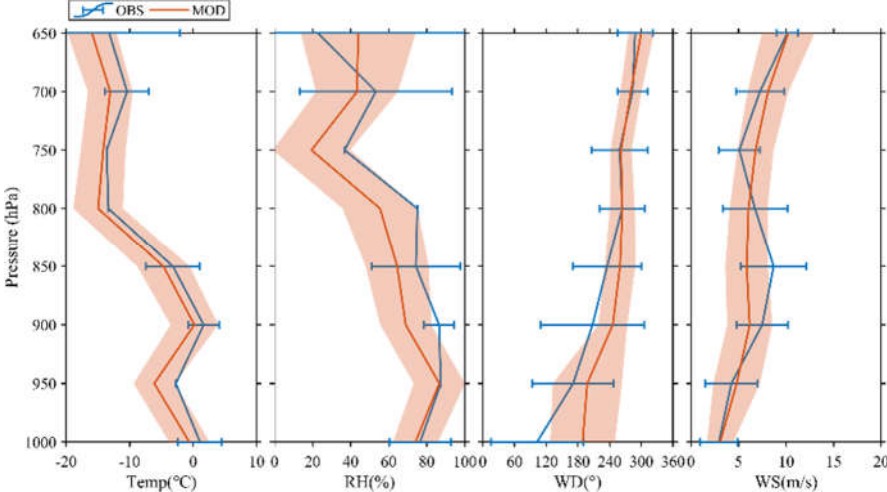

**Figure 5. Comparisons of simulated and observed values of meteorological elements in Beijing in February 2014.**





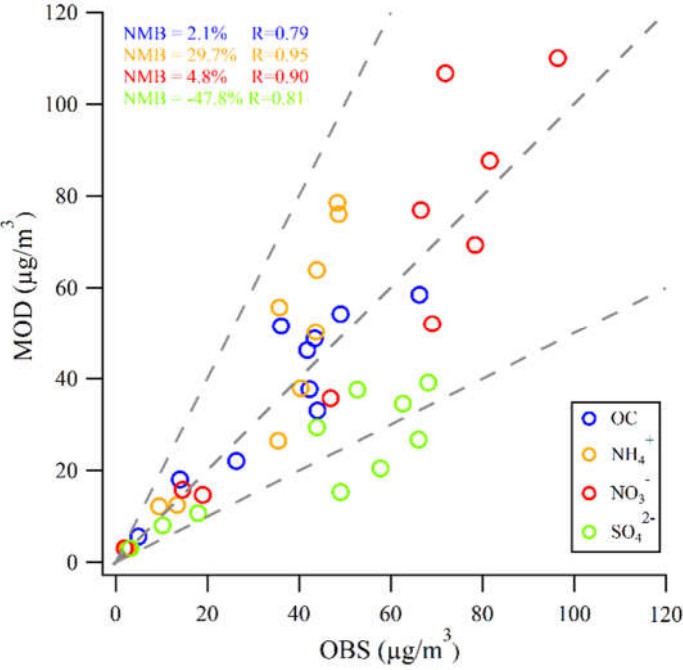

**Figure 6. Evaluation of simulated PM$_{2.5}$ chemical composition concentrations against ground-based observations. The solid line corresponds to the 1:1 line, and the dashed lines correspond to the 1:2 and 2:1 line.**





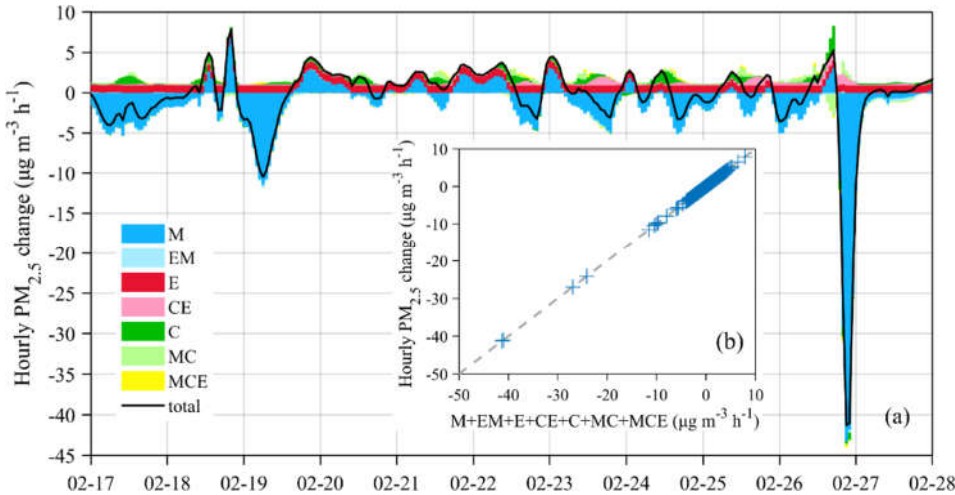

**Figure 7. Decoupling of the PM$_{2.5}$ hourly change (a) and scatter plot (b) of the sum of all contributions versus the hourly change in PM$_{2.5}$. Hourly variation of PM$_{2.5}$ denote the difference in concentration between adjacent hours, and the scattered points all fall on the diagonal line, indicating that the concentration change can be fully resolved.**



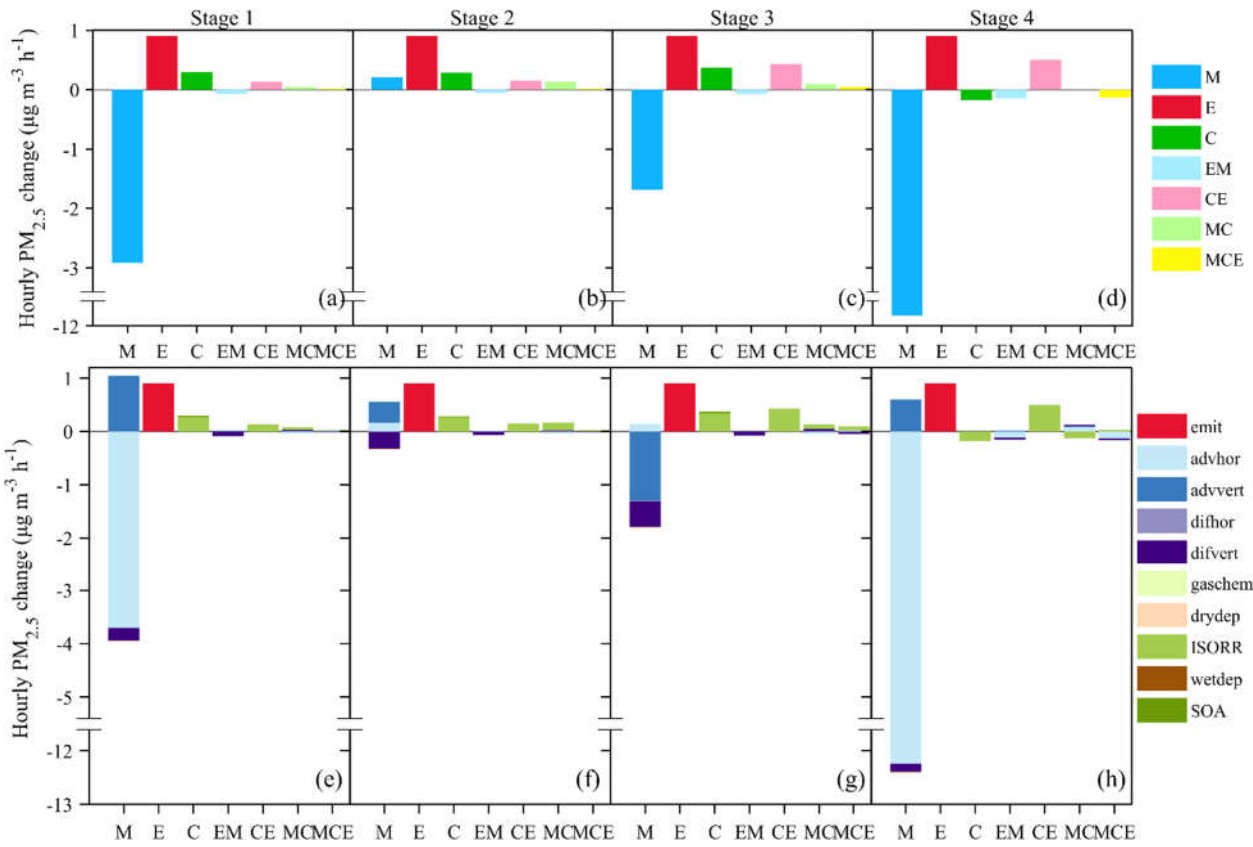

**Figure 8. Averaged hourly contribution of each component in QDA (a-d) and process analysis results for the different components that influence the hourly mean PM$_{2.5}$ change in the total model height at different stages (e-h). (Corresponding values are available in Table 3~4). Take the 'M' bar in (a) for example, the bar of 'M' is composed of six different parts as that displayed in the bar of (e), i.e., 'advhor', 'advvert', 'difhor', 'difvert', 'drydep' and 'wetdep', respectively.**




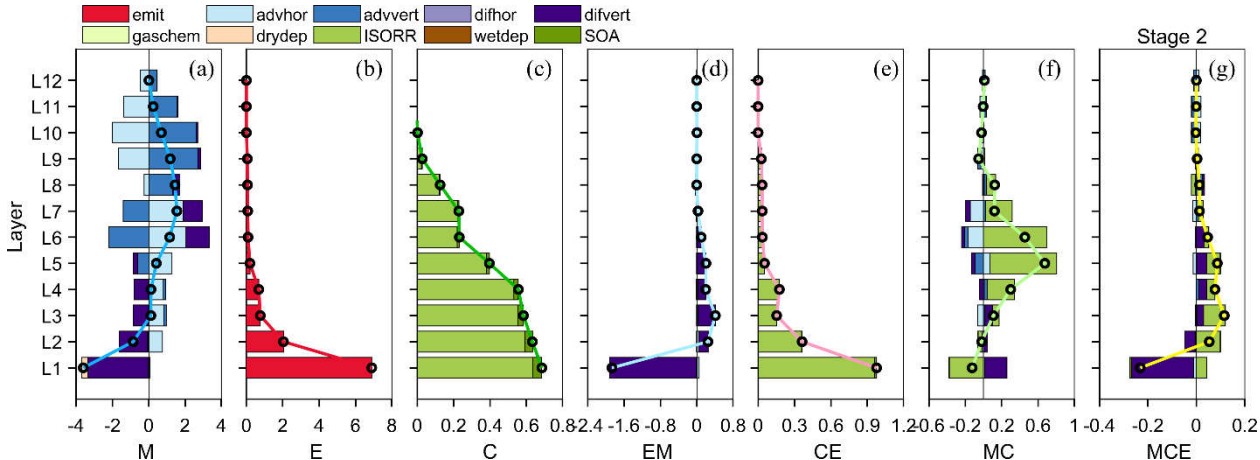

**Figure 9. Vertical process decomposition of the QDA results at stage 2 (the black arrow and colored lines indicate the average change in the PM$_{2.5}$ concentration, results for other stages showed in Fig. S3-S5; unit: μg m$^{-3}$ h$^{-1}$).**



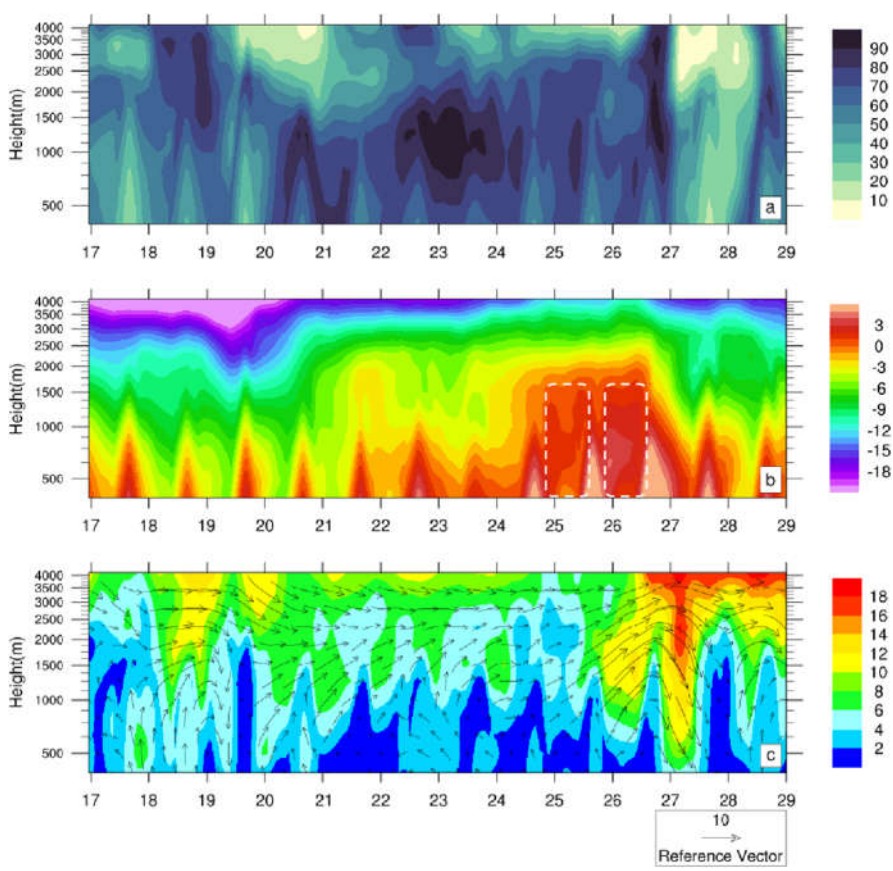

**Figure 10. Vertical distribution of (a)RH, (b) temperature and (c) wind field from 17th Feb to 28th Feb 2014 over Beijing area. The white dotted box in (b) represents a temperature inversion. Vector diagram in (c) represents horizontal wind field, and the shading represents wind speed).**