# Peer review of "A quantitative decoupling analysis (QDA v1.0) method for the assessment of meteorological, emission and chemical contributions to fine particulate pollution"

_Geoscientific Model Development, 2021_

## Author Comment (AC1)

**Response to Referee #1 (gmd-2021-259)**

We Thank Reviewer for his/her constructive comments.

Responses to the comments:

**General comments:** The authors describe a method for determining the contributions of different processes to $PM_{2.5}$ formation and the couplings between the processes. The authors call this method Quantitative Decoupling Analysis (QDA) and apply the method to a haze episode in the Beijing-Tianjin-Hebei region from 17-28 February 2014. The manuscript is generally well written. However, there are three significant problems with the work:

**Comment 1:** The QDA method is not new. This is the Factor Separation method introduced in 1993 and applied in later work. See references below.

**Reply:** Thanks for this important comment. We agree that the QDA method developed in this study share same theoretical basis (Tylor series expansion) with the Factor Separation method introduced by Stein and Alpert (1993), but there are important differences between these two methods. The similarity between the Factor Separation method and the QDA method is that they employ same algorithms to separate the contributions of different factors, while the biggest difference between the Factor Separation method and the QDA method is in the object that they resolved. They are two types of analysis method. The formal is the relative analysis method while the latter is the absolute analysis method. As seen in Fig.R1, the Factor Separation method is designed to resolve the effects of different factors on the differences between model results of two scenarios (i.e., control simulation – base simulation). This makes the contributions of different factors resolved by Factor Separation are relative, which are dependent on the choose of scenarios. It cannot obtain the absolute contribution of different factors. Different from the Factor Separation method, the QDA method aims to track the contributions of different factors on the variations of model results between adjacent timesteps (Fig. R1). This means that the results of QDA methods are absolute, which only depends on the cases we choose and does not need an additional scenario to be compared. Take the estimation of meteorology contribution to the $PM_{2.5}$ concentration as an example. The Factor Separation method need another case with different meteorology condition to estimate the meteorology contribution. This makes the meteorology contribution estimated by Factor Separation method is in fact the contribution induced by meteorology variation between these two cases. It cannot estimate the absolute contribution of meteorology to the variation of $PM_{2.5}$ in an individual case. However, the meteorology contribution estimated by QDA method is the absolute contribution made meteorology processes in this individual case.

Therefore, we believe the QDA method is a new method different from the Factor Separation method. To facility the understanding of the QDA method and highlight its novelty, in the revised manuscript, we have made a more detailed description on the QDA method, including its theoretical basis, algorithms, its realization in model as well as its relationship with the SAA (Factor Separation method) and IPR method,

**1.1 Theoretical basis of the QDA method**

The QDA method is developed based on the Taylor series expansion. Considering that the PM$_{2.5}$ concentration at t step is $PM_{2.5}^t$ and the PM$_{2.5}$ concentration at t+1 step after undergoing the emission, meteorology and chemistry processes with $PM_{2.5}^t$ as initial condition is $PM_{2.5}^{t+1}$, then we could define a function F that denotes the simulated PM$_{2.5}$ concentrations with or without different processes using $PM_{2.5}^t$ as initial concentration, such that:

$$F(0,0,0) = PM_{2.5}^t$$

(R1)

$$F(x_1, x_2, x_3) = PM_{2.5}^{t+1}$$

(R2)

where $F(x_1, x_2, x_3)$ represents the simulated PM$_{2.5}$ concentration with meteorology ($x_1$), emission ($x_2$), and chemistry processes ($x_3$); $F(0,0,0)$ represents the simulated PM$_{2.5}$ concentration without emission, meteorology, and chemistry processes. Therefore, the $PM_{2.5}^t$ and the $PM_{2.5}^{t+1}$ can be seen as the different values of function F with different input data, and the variation of PM$_{2.5}$ concentration between two timesteps can be written as:

$$\Delta PM_{2.5}^{t+1} = PM_{2.5}^{t+1} - PM_{2.5}^t = F(x_1, x_2, x_3) - F(0,0,0)$$

(R3)

where $\Delta PM_{2.5}^{t+1}$ represents the variation of PM$_{2.5}$ concentration from t to t+1 step,

According to Taylor series expansion, the function $F$ can be decomposed as follows:

$$F(x_1, x_2, x_3) - F(0,0,0) = \sum_{i=1}^{3} \frac{\partial F}{\partial x_i} x_i + \frac{1}{2!} \left( \sum_{i=1}^{3} \frac{\partial^2 F}{\partial x_i^2} x_i^2 + 2\frac{\partial^2 F}{\partial x_1 \partial x_2} x_1 x_2 + 2\frac{\partial^2 F}{\partial x_2 \partial x_3} x_2 x_3 + 2\frac{\partial^2 F}{\partial x_1 \partial x_3} x_1 x_3 \right) +$$

$$\frac{1}{3!} \left( \sum_{i=1}^{3} \frac{\partial^3 F}{\partial x_i^3} x_i^3 + \sum_{a=1}^{2} 3\frac{\partial^3 F}{\partial x_1^a \partial x_2^{3-a}} x_1^a x_2^{3-a} + \sum_{a=1}^{2} 3\frac{\partial^3 F}{\partial x_2^a \partial x_3^{3-a}} x_2^a x_3^{3-a} + \sum_{a=1}^{2} 3\frac{\partial^3 F}{\partial x_1^a \partial x_3^{3-a}} x_1^a x_3^{3-a} +$$

$$\frac{\partial^3 F}{\partial x_1 \partial x_2 \partial x_3} 6 x_1 x_2 x_3 \right) + \dots + o^n$$

(R4)

Based on this equation, the terms that only containing a single partial derivative to $x_1$, $x_2$, and $x_3$ (including any higher-order derivatives) are defined as pure contribution of the meteorology (M), emission (E) and chemistry (C) processes to the variation of PM$_{2.5}$ concentrations. Therefore, the term $\frac{\partial F}{\partial x_1} x_1 + \frac{1}{2!} \frac{\partial^2 F}{\partial x_1^2} x_1^2 + \frac{1}{3!} \frac{\partial^3 F}{\partial x_1^3} x_1^3 + \cdots$ in Eq.(R2) is defined as E, term $\frac{\partial F}{\partial x_2} x_2 + \frac{1}{2!} \frac{\partial^2 F}{\partial x_2^2} x_2^2 + \frac{1}{3!} \frac{\partial^3 F}{\partial x_2^3} x_2^3 + \cdots$ is defined as M, and the term $\frac{\partial F}{\partial x_3} x_3 + \frac{1}{2!} \frac{\partial^2 F}{\partial x_3^2} x_3^2 + \frac{1}{3!} \frac{\partial^3 F}{\partial x_3^3} x_3^3 + \cdots$ is defined as C. The cross terms then represent the interaction among different drivers, for example the term $\frac{1}{2!} \frac{2\partial^2 F}{\partial x_1 \partial x_2} x_1 x_2 + \frac{1}{3!} \sum_{a=1}^{2} \frac{3\partial^3 F}{\partial x_1^a \partial x_2^{3-a}} x_1^a x_2^{3-a} + \cdots$ is defined as the interactions between meteorology and emission (ME), the term $\frac{1}{3!} \left( \frac{\partial^3 F}{\partial x_1 \partial x_2 \partial x_3} 6 x_1 x_2 x_3 \right) + \cdots$ is defined as the interactions among emission, meteorology and chemistry processes (MCE). Detailed definitions to the different factors we resolved are available in Table R1. Note that pure here as well as elsewhere in the paper, is in the relative sense meaning that the effect due to one factor is separated

from the other chosen factors. For example, the pure contribution of emission is only due to the direct emission at local space. The variation of PM$_{2.5}$ concentration after emission process are seen as the contribution of meteorology and chemistry. Therefore, the values of E in the QDA method cannot represent the whole effects of emission in the common sense.

According to these definitions, the PM$_{2.5}$ variations from t to t+1 step can be written as the sum of $M^{t+1}$, $E^{t+1}$, $C^{t+1}$, $ME^{t+1}$, $MC^{t+1}$, $CE^{t+1}$, and $MCE^{t+1}$, which is as follows:

$$\Delta PM_{2.5}^{t+1} = M^{t+1} + E^{t+1} + C^{t+1} + ME^{t+1} + MC^{t+1} + CE^{t+1} + MCE^{t+1} \tag{R5}$$

**Table R1 Definition of different factors considers in the QDA method**

| Markers | Equations | Definitions |
|---|---|---|
| M | $\dfrac{\partial F}{\partial x_1}x_1 + \dfrac{1}{2!}\dfrac{\partial^2 F}{\partial x_1^2}x_1^2 + \dfrac{1}{3!}\dfrac{\partial^3 F}{\partial x_1^3}x_1^3 + \cdots$ | Pure contribution of meteorology |
| E | $\dfrac{\partial F}{\partial x_2}x_2 + \dfrac{1}{2!}\dfrac{\partial^2 F}{\partial x_2^2}x_2^2 + \dfrac{1}{3!}\dfrac{\partial^3 F}{\partial x_2^3}x_2^3 + \cdots$ | Pure contribution of emission |
| C | $\dfrac{\partial F}{\partial x_3}x_3 + \dfrac{1}{2!}\dfrac{\partial^2 F}{\partial x_3^2}x_3^2 + \dfrac{1}{3!}\dfrac{\partial^3 F}{\partial x_3^3}x_3^3 + \cdots$ | Pure contribution of chemistry |
| ME | $\dfrac{1}{2!}\dfrac{2\partial^2 F}{\partial x_1 \partial x_2}x_1 x_2 + \dfrac{1}{3!}\sum_{a=1}^{2}\dfrac{3\partial^3 F}{\partial x_1^a \partial x_2^{3-a}}x_1^a x_2^{3-a} + \cdots$ | Coupling contribution of meteorology and emission |
| CE | $\dfrac{1}{2!}\dfrac{2\partial^2 F}{\partial x_2 \partial x_3}x_2 x_3 + \dfrac{1}{3!}\sum_{a=1}^{2}\dfrac{3\partial^3 F}{\partial x_2^a \partial x_3^{3-a}}x_2^a x_3^{3-a} + \cdots$ | Coupling contribution of emission and chemistry |
| MC | $\dfrac{1}{2!}\dfrac{2\partial^2 F}{\partial x_1 \partial x_3}x_1 x_3 + \dfrac{1}{3!}\sum_{a=1}^{2}\dfrac{3\partial^3 F}{\partial x_1^a \partial x_3^{3-a}}x_1^a x_3^{3-a} + \cdots$ | Coupling contribution of meteorology and chemistry |
| MCE | $\dfrac{1}{3!}\left(\dfrac{\partial^3 F}{\partial x_1 \partial x_2 \partial x_3}6x_1 x_2 x_3\right) + \cdots$ | Coupling contribution of emission, meteorology and chemistry |

**1.2 Algorithms of the QDA and its implementation in model**

The QDA method uses a similar algorithms to the factor separation method introduced by Stein and Alpert (1993) to calculated the terms in Eq. (R3). By setting $x_i$ ($i = 1,2,3$) in Eq. (R2) to either 1 or 0, we can simply obtain following equations:

$$F(x_1,0,0) - F(0,0,0) = \frac{\partial F}{\partial x_1}x_1 + \frac{1}{2!}\frac{\partial^2 F}{\partial x_1^2}x_1^2 + \frac{1}{3!}\frac{\partial^3 F}{\partial x_1^3}x_1^3 + \cdots = M$$

(R6)

$$F(0,x_2,0) - F(0,0,0) = \frac{\partial F}{\partial x_2}x_2 + \frac{1}{2!}\frac{\partial^2 F}{\partial x_2^2}x_2^2 + \frac{1}{3!}\frac{\partial^3 F}{\partial x_2^3}x_2^3 + \cdots = E$$

(R7)

$$F(0,0,x_3) - F(0,0,0) = \frac{\partial F}{\partial x_3}x_3 + \frac{1}{2!}\frac{\partial^2 F}{\partial x_3^2}x_3^2 + \frac{1}{3!}\frac{\partial^3 F}{\partial x_3^3}x_3^3 + \cdots = C$$

(R8)

$$F(x_1, x_2, 0) - F(0,0,0) = \frac{\partial F}{\partial x_1} x_1 + \frac{\partial F}{\partial x_2} x_2 + \frac{1}{2!} \left( \frac{\partial^2 F}{\partial x_1^2} x_1^2 + \frac{\partial^2 F}{\partial x_2^2} x_2^2 + 2 \frac{\partial^2 F}{\partial x_1 \partial x_2} x_1 x_2 \right) + \cdots = M + E + ME$$

(R9)

$$F(x_1, 0, x_3) - F(0,0,0) = \frac{\partial F}{\partial x_1} x_1 + \frac{\partial F}{\partial x_3} x_3 + \frac{1}{2!} \left( \frac{\partial^2 F}{\partial x_1^2} x_1^2 + \frac{\partial^2 F}{\partial x_3^2} x_3^2 + 2 \frac{\partial^2 F}{\partial x_1 \partial x_3} x_1 x_3 \right) + \cdots = M + C + MC$$

(R10)

$$F(0, x_2, x_3) - F(0,0,0) = \frac{\partial F}{\partial x_2} x_2 + \frac{\partial F}{\partial x_3} x_3 + \frac{1}{2!} \left( \frac{\partial^2 F}{\partial x_2^2} x_2^2 + \frac{\partial^2 F}{\partial x_3^2} x_3^2 + 2 \frac{\partial^2 F}{\partial x_2 \partial x_3} x_2 x_3 \right) + \cdots = E + C + CE$$

(R11)

where $F(x_1, 0, 0)$, $F(0, x_2, 0)$, $F(0, 0, x_3)$ can be calculated by the simulation that only considers meteorology, emission, and chemistry process from t to t+1 step, respectively (Table R1); $F(x_1, x_2, 0)$, $F(x_1, 0, x_3)$, $F(0, x_2, x_3)$ can be calculated by the simulation that does not including chemistry, emission, and meteorology process from t to t+1 step, respectively. We define these simulations as the accompanying simulation, since their concentrations were updated by the base simulation at each model step as we said in following content. According to Eq. R1 and Eq. R2, the values of $F(0,0,0)$ and $F(x_1, x_2, x_3)$ can be obtained from the base simulation. Based on these equations, each term in Eq. (R3) can be simply calculated by:

$$M^{t+1} = F(x_1, 0, 0)|_{PM_{2.5}^t} - F(0,0,0)|_{PM_{2.5}^t} \tag{R12}$$

$$E^{t+1} = F(0, x_2, 0)|_{PM_{2.5}^t} - F(0,0,0)|_{PM_{2.5}^t} \tag{R13}$$

$$C^{t+1} = F(0, 0, x_3)|_{PM_{2.5}^t} - F(0,0,0)|_{PM_{2.5}^t} \tag{R14}$$

$$ME^{t+1} = F(x_1, x_2, 0)|_{PM_{2.5}^t} - F(x_1, 0, 0)|_{PM_{2.5}^t} - F(0, x_2, 0)|_{PM_{2.5}^t} + F(0,0,0)|_{PM_{2.5}^t} \tag{R15}$$

$$MC^{t+1} = F(x_1, 0, x_3)|_{PM_{2.5}^t} - F(x_1, 0, 0)|_{PM_{2.5}^t} - F(0, 0, x_3)|_{PM_{2.5}^t} + F(0,0,0)|_{PM_{2.5}^t} \tag{R16}$$

$$CE^{t+1} = F(0, x_2, x_3)|_{PM_{2.5}^t} - F(0, x_2, 0)|_{PM_{2.5}^t} - F(0, 0, x_3)|_{PM_{2.5}^t} + F(0,0,0)|_{PM_{2.5}^t} \tag{R17}$$

$$MCE^{t+1} = F(x_1, x_2, x_3)|_{PM_{2.5}^t} + \left( F(x_1, 0, 0)|_{PM_{2.5}^t} + F(0, x_2, 0)|_{PM_{2.5}^t} + F(0, 0, x_3)|_{PM_{2.5}^t} \right) - \left( F(x_1, x_2, 0)|_{PM_{2.5}^t} + \right.$$

$$F(x_1, 0, x_3)|_{PM_{2.5}^t} + F(0, x_2, x_3)|_{PM_{2.5}^t} \left. \right) - F(0,0,0)|_{PM_{2.5}^t} \tag{R18}$$

where $F|_{PM_{2.5}^t}$ denote the simulated PM2.5 concentration with $PM_{2.5}^t$ as the initial condition. Based on Eq. (R1) and Eq. (R2), the values of $F(0,0,0)|_{PM_{2.5}^t}$ and $F(x_1, x_2, x_3)|_{PM_{2.5}^t}$ can be simply obtained from the base simulation, while the other six values are obtained from the results of six accompanying simulations. Since the accompanying simulations at each time step use $PM_{2.5}^t$ as initial condition, the concentrations of PM2.5 and other species in the accompanying simulation will be updated by the base simulation at the start of each model step. For example, the simulation $F(0, 0, x_3)|_{PM_{2.5}^t}$ is run from t to t+1 step without including meteorology processes and emissions

anywhere in the modeling, then a new PM$_{2.5}$ concentration will be obtained from the base simulation for the next time step to drive $F(0,0,x_3)|_{PM_{2.5}^{t+1}}$.

Therefore, in $F(0,0,x_3)$, the meteorology processes and emissions are absent for the entire simulation but the concentrations of PM$_{2.5}$ and other species in each model grid is updated by base simulation in each time step. This enables us to isolate the chemistry and emission and evaluate the contributions of different processes to the variation of PM$_{2.5}$ within a time step. To achieve this, the codes of accompanying simulation were embedded in the code of base simulation so that the simulated results of each accompanying simulation at each time step can be easily and quickly updated.

**Table R2. the descriptions of accompanying simulation in QDA method**

|  | Simulation name | Processes included in the simulations | Target values |
|---|---|---|---|
| Base simulation | S | All physicochemical processes | $F(x_1,x_2,x_3)$, $F(0,0,0)$ |
| Accompanying simulations | S1 | Only meteorological process | $F(x_1,0,0)$ |
|  | S2 | Only emission processes | $F(0,x_2,0)$ |
|  | S3 | Only chemical process | $F(0,0,x_3)$ |
|  | S13 | Meteorological and chemical processes | $F(x_1,0,x_3)$ |
|  | S23 | Emission and chemical processes | $F(0,x_2,x_3)$ |
|  | S12 | Emission and meteorological processes | $F(x_1,x_2,0)$ |

**1.3 Relationship with SAA (Factor Separation) and IPR method**

The scenario analysis approach (SAA) as well as its updated algorithm, Factor Separation method introduced by Stein and Alpert (1993), is an effective tool for performing model sensitivity analysis and for identifying key factors that contribute significantly to model output. Compared with the SAA method, the Factor Separation method is superior in dealing with the nonlinear process that involves two or more factors. By performing multiple sensitivity experiments with different combination of factors, the Factor Separation method allows to assess the impact of a single factor in a nonlinear system as well as the interaction between that factor and others. The similarity between the

Factor Separation method and the QDA method is that they employ same algorithms to separate the contributions of different factors, while the biggest difference between the Factor Separation method and the QDA method is in the object that they resolved. As seen in Fig.R1, the Factor Separation method is designed to resolve the effects of different factors on the differences between model results from two scenarios (i.e., control simulation – base simulation). This makes the contributions of different factors resolved by Factor Separation are in a relative sense, which are dependent on the choose of scenarios. Different from the Factor Separation method, the QDA method aims to track the contributions of different factors on the variations of model results in different time steps (Fig. R1). Therefore, the results of QDA methods are in an absolute sense, which only depends on the cases we choose. In addition, in the Factor Separation method, the sensitivity experiments were run independently with the base simulation, while in the QDA method the sensitivity experiments, i.e., accompanying simulations, are coupled with the base simulation as we illustrate in Sect.1.2.

By analyzing the contribution of each process in the model, the IPR method can be used to resolved the contribution of different physical and chemical processes to the change of pollutant concentration. Considering that the emission process, chemical process and meteorological process are calculated in order in the CTMs, the IPR method is in fact equivalent to one realization of SAA method which calculated the effects of emission, meteorology and chemistry on the variation of model results by conducting three sensitivity simulations (Fig.R1). This makes the IPR method unable to consider the nonlinear effects between different factors, and results in the non-uniqueness of the results of IPR method since we can design different combinations of scenario experiments to calculate the contribution of different factors in the model process. Therefore, although both the IPR method and the QDA method aim to resolve the contribution of different factors to the variation of model results, the QDA method is superior in handling the nonlinearity among different factors through the conduction of more sensitivity simulation.

In all, the QDA method could be seen as a combination of the Factor Separation method and IPR method. It uses the idea of Factor Separation to do the IPR analysis, which for the first time resolve the contributions of different factors as well as their interactions to the variation of model results.

**1.4 Combination with the IPR method**

Since the QDA results only gives the gross effects of emission, meteorology and chemistry processes on the variation of model results, the QDA method is combined with the IPR method to calculate the IPR results for different factors. This is achieved by applying the IPR method to each accompanying simulation. Then the results of different accompanying simulation can be decomposed as follows:

$$F(x_1,0,0)|_{PM_{2.5}^t} = emit_{x1}^{t+1} + advhor_{x1}^{t+1} + advvert_{x1}^{t+1} + difhor_{x1}^{t+1} + difvert_{x1}^{t+1} + wetdep_{x1}^{t+1} + drydep_{x1}^{t+1} +$$

$$gaschem_{x1}^{t+1} + ISORR_{x1}^{t+1} + SOA_{x1}^{t+1} + F(0,0,0)|_{PM_{2.5}^t} \tag{R19}$$

$$F(0,x_2,0)|_{PM_{2.5}^t} = emit_{x2}^{t+1} + advhor_{x2}^{t+1} + advvert_{x2}^{t+1} + difhor_{x2}^{t+1} + difvert_{x2}^{t+1} + wetdep_{x2}^{t+1} + drydep_{x2}^{t+1} +$$

$$gaschem_{x_2}^{t+1} + ISORR_{x_2}^{t+1} + SOA_{x_2}^{t+1} + F(0,0,0)|_{PM_{2.5}^t} \tag{R20}$$

$$F(0,0,x_3)|_{PM_{2.5}^t} = emit_{x_3}^{t+1} + advhor_{x_3}^{t+1} + advvert_{x_3}^{t+1} + difhor_{x_3}^{t+1} + difvert_{x_3}^{t+1} + wetdep_{x_3}^{t+1} + drydep_{x_3}^{t+1} +$$

$$gaschem_{x_3}^{t+1} + ISORR_{x_3}^{t+1} + SOA_{x_3}^{t+1} + F(0,0,0)|_{PM_{2.5}^t} \tag{R21}$$

$$F(x_1,x_2,0)|_{PM_{2.5}^t} = emit_{x_1,x_2}^{t+1} + advhor_{x_1,x_2}^{t+1} + advvert_{x_1,x_2}^{t+1} + difhor_{x_1,x_2}^{t+1} + difvert_{x_1,x_2}^{t+1} + wetdep_{x_1,x_2}^{t+1} +$$

$$drydep_{x_1,x_2}^{t+1} + gaschem_{x_1,x_2}^{t+1} + ISORR_{x_1,x_2}^{t+1} + SOA_{x_1,x_2}^{t+1} + F(0,0,0)|_{PM_{2.5}^t} \tag{R22}$$

$$F(x_1,0,x_3)|_{PM_{2.5}^t} = emit_{x_1,x_3}^{t+1} + advhor_{x_1,x_3}^{t+1} + advvert_{x_1,x_3}^{t+1} + difhor_{x_1,x_3}^{t+1} + difvert_{x_1,x_3}^{t+1} + wetdep_{x_1,x_3}^{t+1} +$$

$$drydep_{x_1,x_3}^{t+1} + gaschem_{x_1,x_3}^{t+1} + ISORR_{x_1,x_3}^{t+1} + SOA_{x_1,x_3}^{t+1} + F(0,0,0)|_{PM_{2.5}^t} \tag{R23}$$

$$F(0,x_2,x_3)|_{PM_{2.5}^t} = emit_{x_2,x_3}^{t+1} + advhor_{x_2,x_3}^{t+1} + advvert_{x_2,x_3}^{t+1} + difhor_{x_2,x_3}^{t+1} + difvert_{x_2,x_3}^{t+1} + wetdep_{x_2,x_3}^{t+1} +$$

$$drydep_{x_2,x_3}^{t+1} + gaschem_{x_2,x_3}^{t+1} + ISORR_{x_2,x_3}^{t+1} + SOA_{x_2,x_3}^{t+1} + F(0,0,0)|_{PM_{2.5}^t} \tag{R24}$$

$$F(x_1,x_2,x_3)|_{PM_{2.5}^t} = emit_{x_1,x_2,x_3}^{t+1} + advhor_{x_1,x_2,x_3}^{t+1} + advvert_{x_1,x_2,x_3}^{t+1} + difhor_{x_1,x_2,x_3}^{t+1} + difvert_{x_1,x_2,x_3}^{t+1} +$$

$$wetdep_{x_1,x_2,x_3}^{t+1} + drydep_{x_1,x_2,x_3}^{t+1} + gaschem_{x_1,x_2,x_3}^{t+1} + ISORR_{x_1,x_2,x_3}^{t+1} + SOA_{x_1,x_2,x_3}^{t+1} + F(0,0,0)|_{PM_{2.5}^t} \tag{R25}$$

where $emit_{x_1}^{t+1}, emit_{x_2}^{t+1}, \cdots, emit_{x_1,x_2,x_3}^{t+1}$ represent the IPR results for emission processes in different accompanying simulation from t to t+1 step, and so do the other processes. Note that some processes in specific accompanying simulation is equal to zero, for example the $emit_{x_1}^{t+1}, gaschem_{x_1}^{t+1}, ISORR_{x_1}^{t+1}$ and $SOA_{x_1}^{t+1}$ term in $F(x_1,0,0)|_{PM_{2.5}^t}$ since $F(x_1,0,0)$ only considers the meteorological processes.

Based on Eq (R19-R25), IPR results for each factor can be calculated as the same way of the contribution of each factor. For example, the formulation of $M^{t+1}$ can be rewritten as follows based on IPR:

$$M^{t+1} = emit_{x_2}^{t+1} + advhor_{x_2}^{t+1} + advvert_{x_2}^{t+1} + difhor_{x_2}^{t+1} + difvert_{x_2}^{t+1} + wetdep_{x_2}^{t+1} + drydep_{x_2}^{t+1} +$$

$$gaschem_{x_2}^{t+1} + ISORR_{x_2}^{t+1} + SOA_{x_2}^{t+1} \tag{R26}$$

Using the same manner, the IPR results for other factors can be calculated according to Eq (R12-R25).

[Figure]

**Figure R1. Comparisons of QDA method with the Factor Separation method and IPR method**

**Comment 2:** The emissions are constant throughout the simulation period, which is not realistic. As a consequence, the contribution of emissions to the $PM_{2.5}$ concentration change is constant throughout the episode, and all the time-variation in the factors and couplings is driven by the meteorology.

**Reply:** Thanks for this important comment. We have updated our results by re-performing the QDA analysis with the considerations of diurnal variation of emissions from different sectors. More analysis on the time variations of the emission contribution was also conducted to provide the decision makers with more intuitive information for the effective control of $PM_{2.5}$ concentrations. Please refer to our responses to Comment 14.

**Comment 3:** The authors consider the influence of three factors on $PM_{2.5}$, total emissions, chemistry, and meteorology, and indicate that their work provides valuable information to decision makers (lines 64-74). $PM_{2.5}$ pollution episodes are driven by anthropogenic emissions and meteorology. Chemistry is a secondary factor that responds to emissions and

meteorology but can be controlled by decision makers only by regulating the anthropogenic emissions. It would be much more relevant to decision makers if the authors had chosen biogenic emissions, anthropogenic emissions, and meteorology as the three factors. Then the full effect of anthropogenic emissions on $PM_{2.5}$ during the episode would be apparent, rather than burying some of the effect in the chemistry factor. Also, the results would help determine if emergency anthropogenic emission controls during episodes would reduce $PM_{2.5}$, which is a goal of the authors' work (lines 71-72) but not a result of their work.

**Reply:** Thanks for this suggestion. We agree with the review that choosing the biogenic emissions, anthropogenic

emissions, and meteorology as three factors would give more apparent results of the full effects of anthropogenic emissions on $PM_{2.5}$, but it cannot distinguish the contributions of primary emission from secondary emissions, which made it difficult to provide the decision makers with insights into the different control strategy of the primary emissions of $PM_{2.5}$ and the emissions of its precursors. Taking the chemistry as factor would help evaluate the contributions of secondary PM2.5, which can give us more insights into the control of the emissions of its precursors. In addition, the QDA method is not only designed to help the decision makers develop emission control strategy but also to help improve air quality models. Taking the chemistry as a factor can help model developers quantitively analyze the influences of different chemistry processes on $PM_{2.5}$ concentrations, this would help the improvement of air quality models. Also, the QDA method developed in this study is targeted at $PM_{2.5}$ concentrations. The biogenic emissions contribute much lower than the anthropogenic emissions to the $PM_{2.5}$ concentrations, thus there is no strong necessity to evaluate the contributions of biogenic emissions. Therefore, we will still use the emission, chemistry, and meteorology as three factors.

**Specific comments:**

**Comment 4:** p. 3, lines 64-72. Decision makers can control anthropogenic emissions and possibly have a minor impact on some biogenic emissions (e.g., types of trees planted in urban areas). Understanding the impact of meteorology on atmospheric concentrations is also important and useful to decision makers. But separating out the impact of chemical reactions does not help regulators reduce atmospheric concentrations. The chemistry factor is controlled by emissions and meteorology, so some (not all) of the chemistry factor represents the impact of emissions. The decision makers need to understand the full impact of the emissions, but that cannot be obtained from the factors that the authors chose.

**Reply:** Thanks for this suggestion. As we illustrate in general comment 3, we think it would be better to choose emission, chemistry, and meteorology as the main factors for the particulate matter pollution.

**Comment 5:** p. 4, lines 89-91. Eq. 2 is incorrect. There should be a factor of 2 in front of the cross terms $\Delta x1 \Delta x2$, $\Delta x2 \Delta x3$, and $\Delta x1 \Delta x3$ in the group of second-order terms and other non-unity factors derived from the binomial coefficients in front of the cross terms in the group of third-order terms. The authors may not have used Eq. 2 and used only Eq. 3. However, if the authors actually used Eq. 2 in their calculations and analyses, they should verify that they used the correct equation, and, if not, the calculations and analyses must be redone. In any case, Eq. 2 should be corrected.

**Reply:** We feel sorry for this mistake. The Eq.2 has been corrected as Eq.(R4) in the revised manuscript. This equation is used to derive Eq.3 in which the terms that only containing a single partial derivative to x1, x2, and x3

(including any higher-order derivatives) are defined as pure effects of M, E and C, for example, the term $\frac{\partial F}{\partial x_1}\Delta x_1 +$

$\frac{1}{2!}\frac{\partial^2 F}{\partial x_1^2}\Delta x_1^2 + \frac{1}{3!}\frac{\partial^3 F}{\partial x_1^3}\Delta x_1^3 + \cdots$ in Eq.2 is defined as the pure effects of M. The cross terms represent the interaction among

the different drivers, for example the $\frac{1}{2!}\frac{2\partial^2 F}{\partial x_1 \partial x_2}\Delta x_1 \Delta x_2 + \frac{1}{3!}\sum_{a=1}^{2}\frac{3\partial^3 F}{\partial x_1^a \partial x_2^{3-a}}\Delta x_1^a \Delta x_2^{3-a} + \cdots$ is defined as the interactions

between meteorology and emission. According to this definition, the Eq.2 can be rewritten as Eq.3, which in fact serve

as the theoretical basis for the decomposition of the PM$_{2.5}$ changes into the contributions of M, E, C, ME, CE, MC and

MCE. Thus, although we did not use Eq.2 in the calculation and analysis, it is necessary to retain the Eq.2 in the

context. Detailed information is available our responses to Comment 1.

**Comment 6:** p. 4, line 104. The interaction between emissions and meteorology is bi-directional. Higher temperatures

increase evaporative emissions from gasoline vehicles, higher temperatures and greater sunlight increase isoprene

emissions from plants, etc.

**Reply:** Thanks for this comment. We have revised this sentence as "Note that although the effects of emissions and

meteorology is bi-directional, for example the higher temperatures would increase evaporative emissions from

gasoline vehicles, the effect of emissions on meteorology is unidirectional in our application since we did not have an

online emission model to represent the interactions between emissions and meteorology, which would be a limitation

of our work."

**Comment 7:** p.4, lines 106-112 and Table 1. There should be a more complete description of the simulations and

what is different between simulation M1 and the other simulations. In particular, for simulation M4, are the

meteorological processes and emissions absent for the entire simulation? If so, what PM$_{2.5}$ could there be in a grid cell

other than the initial PM$_{2.5}$ concentration, which is stationary in space because no meteorological processes are

included? Is the PM$_{2.5}$ concentration at the start of a specific time step taken from simulation M1, simulation M4 is

run over that time step without including meteorology processes or emissions anywhere in the modeling domain, then

a new PM$_{2.5}$ concentration is obtained from simulation M1 for the next time step? What the authors did is very

unclear.

**Reply:** Thanks for this important comment. Yes, the simulation M2 – M7 are all accompanying simulations that the

concentrations of PM$_{2.5}$ and other species at the start of each model step is taken from simulation M1. Thus, as the

review said, the simulation M4 is run over that time step without including meteorology processes and emissions

anywhere in the modeling, and a new PM$_{2.5}$ concentration is obtained from simulation M1 for the next time step. So,

in M4, the meteorology processes and emissions are absent for the entire simulation but the concentrations of PM$_{2.5}$

and other species in each model grid is updated by M1 in each time step. This is in fact a key difference between our

method and the Factor Separation method, which enable us to evaluate the contributions of different processes to the

variation of PM$_{2.5}$ within a time step. Following the suggestions of reviewer, we have added more complete

description of the accompanying simulation in the revised manuscript and clarify the differences between the QDA method and Factor Separation method (Please see our responses to Comment 1).

**Comment 8:** p. 4, lines 111-114. Did the authors run the base simulation 6 times, each time with one of the "accompanying" simulations? That would effectively be 12 simulations. A simpler approach would seem to be running the base simulation once, recording the timesteps used, and then running each of the "accompanying" simulations once with the same timesteps used for the base case. That would reduce the number of simulations needed to 7.

**Reply:** Since we integrate the code of accompanying simulation into the code of base simulation, the accompanying simulation is running simultaneously with the base simulation. Therefore, as the reviewer said, the total number of simulation is 7.

**Comment 9:** p. 4, lines 119-120. There should be a detailed explanation of how IPR is applied to the results for each factor. It is unclear how this was done. Simulation M4 (C factor) does not contain emissions, so the chemistry will be different from that when emissions are present. Are the IPR results then meaningful for C?

**Reply:** Thanks for this suggestion. The IPR results for different factors is calculated by applying the IPR to each accompanying simulation. Please refer to our response to Comment 1 for more detailed description for the application of the IPR to the QDA results.

**Comment 10:** pp. 3-5. The QDA method is not new. This is the Factor Separation method introduced by U. Stein and P. Alpert, Factor separation in numerical simulations, J. Atmos. Sci. 50, 2107-2115 (1993). Subsequently, Tao et al. applied the method to separate the contributions of area, mobile, and point source emissions to ozone and their interactions (Tao et al.,    , Atmos. Environ. 39, 1869-1877 (2005)). The authors should not refer to QDA as a new method and should credit Stein and Alpert and Tao et al. by including their references in the manuscript.

**Reply:** Thanks for this important comment. We agree that the QDA method developed in this study share same theoretical basis (Tylor series expansion) with the Factor Separation method introduced by Stein and Alpert (1993), but we believe it is a new method different from the Factor Separation method. In Tao et al. (2005), the Factor Separation method was used to assess the contribution of area, mobile, and point source emissions to the ozone. However, what Tao et al. (2005) resolved is the differences in ozone concentrations under different emission conditions. They did not estimate the contributions of emission to the variation of $O_3$ concentrations between adjacent timestep. The sum of emission contribution made by different sources estimated by Tao et al. (2005) was not equal to the differences of $O_3$ concentration between adjacent timestep. Please refer to our responses to Comment 1 for detailed comparisons between QDA method and Factor Separation method.

**Comment 11:** p. 7, line 187. What is MBE? This is not defined in Table S1. MB is -13.7 µg/m3 and ME is 42.1 µg/m3 (Table S2) so it cannot be either of those two statistics.

**Reply:** We feel sorry for this confusion. The MBE is mean bias error which is the same as MB (mean bias). We have corrected this error in the revised manuscript.

**Comment 12:** p. 7, line 191. Again, what is MBE and where are these values (7.1 and 5.3 µg/m3 ) in Table S2? If the values are discussed in the manuscript, they should be in Table S2.

**Reply:** We feel sorry for this confusion. The MBE is mean bias error which is the same as MB (mean bias). We have corrected this error in the revised manuscript.

**Comment 13:** p. 7, line 194. There is a more recent paper (L. Huang et al., Atmos. Chem. Phys. 21, 2725-2743 (2021)) that gives goals and criteria specifically for PM$_{2.5}$ simulations in China.

**Reply:** Thanks for this suggestion. According to the results from Huang et al. (2021), the simulated total PM$_{2.5}$ concentrations all satisfied the normalized mean bias (NMB), normalized mean error (NBE), correlation coefficients (R), and index of agreement (IOA) performance standards (NMB<20%, NBE<45%, R<0.6 and IOA > 0.7). Following the suggestion of reviewer, the goals and criteria proposed by L.Huang et al., (2021) are used to illuminate the robustness

and reliability of out mode results in the revised manuscript.

**Comment 14:** p. 7, lines 210-212. For their analyses, the authors fixed the emissions to be constant in time. It is unclear why this is necessary for the method, and it is a serious limitation of their work. Neither the anthropogenic nor the biogenic emissions are constant in time; there are large variations over the diurnal cycle. As a consequence of the authors' assumption of constant emissions, their calculated emission contribution is constant over all 12 days of the episode (Figures 7 and 8, Tables 3 and 4). This is not an interesting or very valuable result, especially for the decision makers/regulators. We cannot control the meteorology, only the anthropogenic emissions, so the important question is to what extent instituting greater emission controls during stagnation events will improve air quality. The authors' results do not provide any insight on that question. Further, the assumption of constant emissions also influences the chemical contribution because time-varying emissions would very likely give much greater variation in the chemistry contribution.

**Reply:** Thanks for this comment. Since the bottom-up emissions are only available at monthly resolution and it is difficult to accurately estimate the time-variation of emissions from different sectors, we fixed the emission to be constant in time during simulation which would lead to uncertainty in our QDA results. However, to fix the emission to be constant in time is not necessary for the QDA method. Following the suggestions of the reviewer, we

re-performed the QDA analysis with the considerations of diurnal variation of emissions from different sectors. Figure R2 shows the diurnal profile of the emissions from different sectors obtained from the MIX inventory(Li et al., 2017), which generally shows higher emissions during the daytime than the nighttime. The transport and residential emission also show a double-peak pattern in their diurnal profile.

[Figure]

Figure R2 the diurnal profile of emissions from different sectors

Figure R3 shows the updated time series of the calculated contributions of emission, meteorology, chemistry and their interactions to the PM$_{2.5}$ variations. Compared with the QDA results without considerations of emission variation, the time-variation of the updated QDA results is generally larger. For example, the calculated meteorological contributions (M) ranges from -48.7 to 7.4 $\mu g \cdot m^{-3} \cdot h^{-1}$ when the emission variation was considered, larger than the values of M (-42–8 $\mu g \cdot m^{-3} \cdot h^{-1}$) without the consideration of emission variation. The time-varying emission also induces larger variation in the contribution of the coupling effects of emission and chemistry, with the calculated contribution of EC ranging from 0 to 1.8 $\mu g \cdot m^{-3} \cdot h^{-1}$ higher than the values of EC (0.1–1.3 $\mu g \cdot m^{-3} \cdot h^{-1}$) without consideration of emission variation.

[Figure]

**Figure R3 time series of hourly PM₂.₅ variations between adjacent hours (black lines) from 17 Feb to 28 Feb, 2014 as well as the contributions of**

Figure R4 gives more detailed results for the diurnal variation of the contributions of different factors during the first three stage of episode over the Beijing area. The last stage is not analyzed since it did not last for one day. According to fig. R4a, the PM₂.₅ concentration decreased by 14.3 µg m−3 during the period of Stage 1 (from 91.2 µg m−3 at 00:00 LST to 76.9 µg m−3 at 23:00 LST). But in Stage 2 (fig. R4b), the PM₂.₅ concentration exhibits a significant monotonic growth, with a daily increment of 37.1 µg m$^{-3}$ (from 56.7 µg m$^{-3}$ at 00:00 LST to 93.8 µg m$^{-3}$ at 23:00 LST). The diurnal variation of PM₂.₅ is small in Stage 3 (fig. R4c), only increased by about 6.3 µg m$^{-3}$. This indicates that Stage 2 has the most favorable environmental conditions for the growth of PM₂.₅, leading to the most significant change of PM₂.₅ concentration compared to other stages. The daily concentration changes in Stage 1 and Stage 3 are both small, indicates that the environment in these periods tend to maintain the stability of PM₂.₅.

The QDA results suggest that the contribution of pure meteorology contribution (M) was generally negative during the first stage, especially at forenoon (05:00–8:00 LST) and afternoon (15:00–17:00 LST) with estimated values of M up to -3 $\mu g \cdot m^{-3} \cdot h^{-1}$. The scavenging effects of M almost become zero during 12:00–15:00 LST. In addition, the contribution of the interaction between meteorology and chemistry become larger, together with the larger pure contribution of emission (E) and chemistry (C), making the PM₂.₅ concentration increased slightly during that time.

However, the values of M turned to be positive during most time of stage 2 especially during the nighttime (fig.R3e), with estimated values of M up to 2.2 $\mu g \cdot m^{-3} \cdot h^{-1}$, much higher than the values of E and C. This suggest that the meteorology dominated the increases of PM₂.₅ at the nighttime of stage 2, and that the control of local emission, with the values of E only ranging from 0.3 to 0.7 $\mu g \cdot m^{-3} \cdot h^{-1}$ during nighttime, may only has little

effects on the control of PM$_{2.5}$. However, the meteorology contribution contains the contribution of transportation of non-local PM$_{2.5}$ concentrations, thus it should be more effective to control the emissions outside Beijing during stage 2, which would effectively slow down the accumulation of PM$_{2.5}$ and may prevent the occurrence of potential heavy haze episode. Although the pure contribution of meteorology (M) become negative during 12:00–18:00 LST, ranging from -1.6 to -0.1 $\mu g \cdot m^{-3} \cdot h^{-1}$, the coupling effects between meteorology and chemical (MC) become positive during that time (0.1–0.6 $\mu g \cdot m^{-3} \cdot h^{-1}$), which indicates that the meteorology condition favors the chemical production of PM$_{2.5}$. The pure contribution of emission and chemistry also become positive and together with MC counteract the scavenging effects of meteorology. This suggests that local emission control both for PM$_{2.5}$ and its precursors is needed if we aimed to migrate the PM$_{2.5}$ pollution at this time.

At stage 3 (fig. R4f), the concentration of PM$_{2.5}$ was maintained at a high level with small fluctuation, which indicates that the contributions of different factors generally reach an equilibrium. The pure contribution of meteorology was relatively weak during the nighttime, but indicates significant scavenging effects during 13:00–19:00 LST with the values of M ranging from -5.1 to -2.7 $\mu g \cdot m^{-3} \cdot h^{-1}$. However, the values of E and CE also increased significantly during that time especially for CE, with maximum values up to 1.2 and 1.6 $\mu g \cdot m^{-3} \cdot h^{-1}$, respectively, which counteract the negative contribution of meteorology. As a result, the PM$_{2.5}$ concentration only slightly decreased during that time. This suggests that for this case, it is still necessary to control the local emissions of PM$_{2.5}$ and its precursor at stage 3.

It is worth noting that the E only contains the contributions of direct emission of PM$_{2.5}$ at local space, thus the values of E kept the same during the three stages which only represents the diurnal variation of emission. However, the contributions of emission in the common sense should also include the coupling effects between the emission and other factors. For example, the values of CE were large at stage3, thus the implementation of emission reduction during that stage would have greater effects on the PM$_{2.5}$ reduction than those indicates by the pure contribution of emission. For the contributions of chemical, it provides us with insights into the control of the emissions of precursors. Also, the pure contribution of Meteorology contains the contributions of transportation of non-local PM$_{2.5}$ concentrations, thus it can serve as an indictor of the contributions of non-local emissions. Therefore, as we illustrate above, the QDA results can provide quantitative information on the relative importance of the effects of scavenging effects of meteorology, local emission, non-local emission and precursors' emission on the PM$_{2.5}$ concentration, which could provide valuable insight on the development of emission control strategy during different stages of episode.

[Figure]

Figure R4. Diurnal variation of the vertical average concentrations of PM$_{2.5}$ as well as its compositions (a-c), and that of the contributions of different factors (d-f) as well as different meteorological parameters (g-i) during the first three stage.

**Comment 15:** p. 8, lines 234-239. These conclusions are well-known from many previous studies.

**Reply:** Thanks for this comment. The QDA results can provide us the quantitative results of the contributions of meteorology, chemical, emission as well as their interactions on the variation of PM$_{2.5}$, which has not been given by previous studies to our best knowledge. We have rewritten this part to highlight the contributions of our work.

**Comment 16:** p. 9, line 276. It is unclear what the range of -0.86 to 1.86 represents. It is much wider than what the results in Table 3 suggest.

**Reply:** We feel sorry for this confusion. The range of -0.86 to 1.86 here denote the range of total coupling effects (i.e., CE +EM+MCE).

**Comment 17:** pp. 10-11, Section 3.4. Many of the conclusions here are well-known from prior work, and the Factor Separation method (QDA) adds little new information to the prior work. At most, this section shows consistency between the Factor Separation method and the results of previous studies, but there is no detailed evaluation of the Factor Separation method.

**Reply:** As we replied in comment 1. The QDA method is different from the Factor Separation method. This section is used to evaluate the results of QDA analysis, which is necessary for helping the potential users or readers understand the robustness and reliability of the QDA method.

**Comment 18:** p. 10, lines 281-282. Yes, the results in the paper do not give much information about the

importance of emissions and therefore are not of much use to decision makers.

**Reply:** Thanks for this comment. It is worth noting that the E only contains the contributions of direct emission of $PM_{2.5}$ at local space. It neither contains the contributions of transboundary transportation of emission nor the contributions of precursors' emissions. Therefore, the values of E cannot represent the whole effects of emission in the common sense. Instead, a comprehensive analysis on the QDA results should be conducted to assess the whole effects of emission. As we illustrated in Comment 14, the QDA results can provide quantitative information on the relative importance of the effects of scavenging effects of meteorology, local emission, non-local emission and precursors' emission on the $PM_{2.5}$ concentration, which could provide valuable insight on the development of emission control strategy during different stages of episode. To highlight the importance of the QDA method for the decision makers, we have added a new section named *suggestions for the decision makers* in the revised manuscript.

**Comment 19:** pp. 11-12. Again, QDA is not a new method and most of the conclusions here are not new.

**Reply:** Thanks for this comment. The QDA is a new method different from the Factor Separation method (Please see our responses to Comment 1) which for the first time give the quantitative analysis of the contribution of emission, meteorology, chemistry as well as their interactions to the variation of $PM_{2.5}$. we also revised the conclusion part in the revised manuscript to highlight the contributions of our work.

**Technical corrections**

**Comment 20:** p. 2, lines 54-55. "However, due to the nonrepeatability of individual pollution cases, … ." Not clear what is meant here. If one has an estimate of the meteorological fields from a weather model and an estimate of the emissions, the air quality model can estimate the atmospheric concentration of PM for days in different years ("individual cases"). Not clear why sensitivity experiments are necessary to "fully reproduce the individual cases."

**Reply:** We feel sorry for this confusion. Here we mean that for the SAA method or Factor Separation method, we cannot find an appropriate meteorology condition to estimate the contributions of meteorology in individual pollution cases. This makes the contributions of different factors resolved by Factor Separation are in a relative sense, which are dependent on the choose of scenarios.

**Comment 21:** p. 2, line 57. Define the PLMA acronym.

**Reply:** Done. The PLMA denotes the parameters linking air-quality to meteorological elements.

**Comment 22:** p. 5, line 133. Should be "nitrate" not "nitrite"?

**Reply:** Done.

**Comment 23:** p. 5, line 149. Was it MOZART v 2.4 or v 2.5?

**Reply:** we feel sorry for this mistake, it is MOZART v2.5.

**Comment 24:** p. 6, line 159. "The" should be "the"

**Reply:** Done.

**Comment 25:** Figure 2 (b). The legend should be larger.

**Reply:** Done. We have enlarged the legend of Figure 2(b) in the revised manuscript.

**Comment 26:** p. 6, lines 179-180. It would be clearer to use the same nomenclature for these statistics as in Table S1. Table S1 has the conventional names.

**Reply:** Done.

**Comment 27:** Figure 6 caption. There is no solid line in the figure, only 3 dashed lines. Do the points represent 24 hour averages? For which days?

**Reply:** We feel sorry for this confusion. We have revised the Figure 6 in the revised manuscript. yes, the points represent the 24-hour averages of $PM_{2.5}$ chemical composition concentration from17 Feb to 28 Feb, 2014.

**Comment 28:** Tables 3 and 4 should be in the Supporting Information because they repeat the information in Figure 8.

**Reply:** Done.

**Comment 29:** p. 9, line 259. Not clear why one limit is -3%. This seems to be a comparison of the magnitudes of the two quantities, in which case the limit would be +3%.

**Reply:** Here the quantity denotes the ration of total coupling effect (COUP) to the $PM_{2.5}$ variation. Thus, -3% denotes that the COUP has an opposite sign to the $PM_{2.5}$ variation.

**Comment 30:** p. 10, line 293. Should it be "from stages 1 to 2" instead of: from stages 2 to 1"?

**Reply:** Done.

**Comment 31:** p. 11, line 311. Define the acronym PLAM.

**Reply: Done.** The PLMA denotes the parameters linking air-quality to meteorological elements.

**Comment 32:** p. 14, lines 436-437. The title of the paper should not be in all capitals.

**Reply:** Done.

**References:**

Huang, L., Zhu, Y., Zhai, H., Xue, S., Zhu, T., Shao, Y., Liu, Z., Emery, C., Yarwood, G., Wang, Y., Fu, J., Zhang, K., and Li, L.: Recommendations on benchmarks for numerical air quality model applications in China – Part 1: PM2.5 and chemical species, Atmos. Chem. Phys., 21, 2725-2743, 10.5194/acp-21-2725-2021, 2021.

Li, M., Zhang, Q., Kurokawa, J. I., Woo, J. H., He, K., Lu, Z., Ohara, T., Song, Y., Streets, D. G., Carmichael, G. R., Cheng, Y., Hong, C., Huo, H., Jiang, X., Kang, S., Liu, F., Su, H., and Zheng, B.: MIX: a mosaic Asian anthropogenic emission inventory under the international collaboration framework of the MICS-Asia and HTAP, Atmos. Chem. Phys., 17, 935-963, 10.5194/acp-17-935-2017, 2017.

Stein, U. and Alpert, P.: FACTOR SEPARATION IN NUMERICAL SIMULATIONS, Journal of the Atmospheric Sciences, 50, 2107-2115, 10.1175/1520-0469(1993)050<2107:Fsins>2.0.Co;2, 1993.

Tao, Z. N., Larson, S. M., Williams, A., Caughey, M., and Wuebbles, D. J.: Area, mobile, and point source contributions to ground level ozone: a summer simulation across the continental USA, Atmos. Environ., 39, 1869-1877, 10.1016/j.atmosenv.2004.12.001, 2005.

---

## Author Comment (AC2)

**Response to Referee #2 (gmd-2021-259)**

We Thank Reviewer for his/her constructive comments.

Responses to the comments:

The manuscript 'A quantitative decoupling analysis (QDA v1.0) method for the assessment of meteorological, emission and chemical contributions to fine particulate pollution' written by Junhua Wang presented the QDA method as novel way to evaluate meteorology, emission, and chemistry processes involved for the aerosol formation. Although the concept of this method is interesting, I cannot fully understand the description of method itself and therefore go through to result and discussion section well. At the current presentation quality, this manuscript cannot be considered for publication. At this round, I would like to reject this manuscript. Before considering the possible publication, I sincerely request the fundamental amendments. I wish the following major and minor comments will help to revise this manuscript. **Reply:** The authors appreciate the reviewer for his/her constructive and up-to-point comments. We have carefully considered the comments and revised the manuscript accordingly. Please refer to our responses for more details given below.

**Major comments:**

**Comment 1:** The description of QDA and its relation to IPR. The newly developed QDA method is just the using of six accompanying simulations to calculate M, E, and C terms. In this sense, for example, to drive M term, this method seems to be identical to the SAA as described in the introduction. Actually, how to conduct six accompanying simulations is unclear. Under each time-step simulation, how can do the base-model derive each process? The detailed description of M2-M7 is required to understand the QDA method. In addition, without E term, C term cannot be driven due to the absence of precursors. Therefore, I guess that EC term inherently connected, and it could be hard to be divided. Moreover, on P4, L118, it was stated that "The above QDA method can also be combine with the IPR method to resolve more detailed information...". This statement is confusing to me because this impressed that QDA is just the using of IPR. Under the current presentation quality, it is difficult to understand QDA method and I cannot recognize this method as novel way in modeling analysis.

**Reply:** We feel sorry that we did not provide enough description on the QDA method. We have made a more detailed description on the QDA method, including its theoretical basis, algorithms, its realization in model as well as its relationship with the SAA (Factor Separation method) and IPR method, to facilitate the understanding of the QDA method and highlight the novelty of QDA method.

**1.1 Theoretical basis of the QDA method**

The QDA method is developed based on the Taylor series expansion. Considering that the PM2.5 concentration at t step is  $PM_{2.5}^t$  and the PM2.5 concentration at t+1 step after undergoing the emission, meteorology and chemistry processes with  $PM_{2.5}^t$  as initial condition is  $PM_{2.5}^{t+1}$ , then we could define a function F that denotes the simulated PM2.5

concentrations with or without different processes using  $PM_{2.5}^t$  as initial concentration, such that:

$$F(0,0,0) = PM_{2.5}^t \tag{R1}$$

$$F(x_1, x_2, x_3) = PM_{2.5}^{t+1}$$
(R2)

where  $F(x_1, x_2, x_3)$  represents the simulated PM2.5 concentration with meteorology  $(x_1)$ , emission  $(x_2)$ , and chemistry processes  $(x_3)$ ; F(0,0,0) represents the simulated PM2.5 concentration without emission, meteorology, and chemistry processes. Therefore, the  $PM_{2.5}^t$  and the  $PM_{2.5}^{t+1}$  can be seen as the different values of function F with different input data, and the variation of PM2.5 concentration between two timesteps can be written as:

$$\Delta PM_{2.5}^{t+1} = PM_{2.5}^{t+1} - PM_{2.5}^{t} = F(x_1, x_2, x_3) - F(0, 0, 0)$$
(R3)

where  $\Delta PM_{2.5}^{t+1}$  represents the variation of PM2.5 concentration from t to t+1 step,

According to Taylor series expansion, the function F can be decomposed as follows:

$$F(x_{1}, x_{2}, x_{3}) - F(0, 0, 0) = \sum_{i=1}^{3} \frac{\partial F}{\partial x_{i}} x_{i} + \frac{1}{2!} \left( \sum_{i=1}^{3} \frac{\partial^{2}F}{\partial x_{i}^{2}} x_{i}^{2} + 2 \frac{\partial^{2}F}{\partial x_{1}\partial x_{2}} x_{1} x_{2} + 2 \frac{\partial^{2}F}{\partial x_{2}\partial x_{3}} x_{2} x_{3} + 2 \frac{\partial^{2}F}{\partial x_{1}\partial x_{3}} x_{1} x_{3} \right) + \frac{1}{3!} \left( \sum_{i=1}^{3} \frac{\partial^{3}F}{\partial x_{i}^{3}} x_{i}^{3} + \sum_{a=1}^{2} 3 \frac{\partial^{3}F}{\partial x_{1}^{a}\partial x_{2}^{3-a}} x_{1}^{a} x_{2}^{3-a} + \sum_{a=1}^{2} 3 \frac{\partial^{3}F}{\partial x_{2}^{a}\partial x_{3}^{3-a}} x_{2}^{a} x_{3}^{3-a} + \sum_{a=1}^{2} 3 \frac{\partial^{3}F}{\partial x_{1}^{a}\partial x_{3}^{3-a}} x_{1}^{a} x_{3}^{3-a} + \frac{\partial^{3}F}{\partial x_{1}^{a}\partial x_{2}^{3-a}} x_{1}^{a} x_{2}^{3-a} + \sum_{a=1}^{2} 3 \frac{\partial^{3}F}{\partial x_{2}^{a}\partial x_{3}^{3-a}} x_{1}^{a} x_{3}^{3-a} + \frac{\partial^{3}F}{\partial x_{1}^{a}\partial x_{2}^{3-a}} x_{1}^{a} x_{3}^{3-a} + \sum_{a=1}^{2} 3 \frac{\partial^{3}F}{\partial x_{1}^{a}\partial x_{3}^{3-a}} x_{1}^{a} x_{3}^{3-a} + \frac{\partial^{3}F}{\partial x_{1}^{a}\partial x_{3}^{3-a}} x_{1}^{a} x_{3}^{3-a} + \frac{\partial^{3}F}{\partial x_{1}^{a}\partial x_{2}^{3-a}} x_{3}^{a} x_{3}^{3-a} + \frac{\partial^{3}F}{\partial x_{1}^{a}\partial x_{3}^{3-a}} x_{1}^{a} x_{3}^{3-a} + \frac{\partial^{3}F}{\partial x_{1}^{a}\partial x_{2}^{3-a}} x_{3}^{a} x_{3}^{a-a} + \frac{\partial^{3}F}{\partial x_{1}^{a}\partial x_{3}^{3-a}} x_{3}^{a-a} + \frac{\partial^{3}F}{\partial x_{1}^{a}\partial x_{2}^{3-a}} x_{3}^{a-a} + \frac{\partial^{3}F}{\partial x_{1}^{a}\partial x_{3}^{3-a}} x_{3}^{a-a} + \frac{\partial^{3}F}{\partial x_{1}^{a}\partial x_{3}^{a-a}} x_{3}^{a-a} + \frac{\partial^{3}F}{\partial x_{1}^{a}\partial x_{3}^{3-a}} x_{3}^{a-a} + \frac{\partial^{3}F}{\partial x_{1}^{a}\partial x_{3}^{3-a}} + \frac{\partial^{3}F}{\partial x_{1}^{a}\partial x_{3}^{a-a}} + \frac{\partial^{3}F}{\partial x_{1}$$

Based on this equation, the terms that only containing a single partial derivative to  $x_1$ ,  $x_2$ , and $x_3$  (including any higherorder derivatives) are defined as pure contribution of the meteorology (M), emission (E) and chemistry (C) processes to the variation of PM2.5 concentrations. Therefore, the term  $\frac{\partial F}{\partial x_1}x_1 + \frac{1}{2!}\frac{\partial^2 F}{\partial x_1^2}x_1^2 + \frac{1}{3!}\frac{\partial^3 F}{\partial x_1^2}x_1^3 + \cdots$  in Eq.(R2) is defined as E, term  $\frac{\partial F}{\partial x_2}x_2 + \frac{1}{2!}\frac{\partial^2 F}{\partial x_2^2}x_2^2 + \frac{1}{3!}\frac{\partial^3 F}{\partial x_2^2}x_2^3 + \cdots$  is defined as M, and the term  $\frac{\partial F}{\partial x_3}x_3 + \frac{1}{2!}\frac{\partial^2 F}{\partial x_3^2}x_3^2 + \frac{1}{3!}\frac{\partial^3 F}{\partial x_3^2}x_3^3 + \cdots$  is defined as C. The cross terms then represent the interaction among different drivers, for example the term  $\frac{1}{2!}\frac{2\partial^2 F}{\partial x_1 \partial x_2}x_1x_2 + \frac{1}{3!}\sum_{l=1}^{2}\frac{2\partial^3 F}{\partial x_1^2 \partial x_2^{2-a}}x_1^a x_2^{3-a} + \cdots$  is defined as the interactions between meteorology and emission (ME), the term  $\frac{1}{3!}\left(\frac{\partial^3 F}{\partial x_1 \partial x_2 \partial x_3} 6x_1x_2x_3\right) + \cdots$  is defined as the interactions among emission, meteorology and chemistry processes (MCE). Detailed definitions to the different factors we resolved are available in Table R1. Note that pure here as well as elsewhere in the paper, is in the relative sense meaning that the effect due to one factor is separated from the other chosen factors. For example, the pure contribution of emission is only due to the direct emission at local space. The variation of PM2.5 concentration after emission process are seen as the contribution of meteorology and chemistry. Therefore, the values of E in the QDA method cannot represent the whole effects of emission in the common sense.

According to these definitions, the PM2.5 variations from t to t+1 step can be written as the sum of  $M^{t+1}$ ,  $E^{t+1}$ ,  $C^{t+1}$ ,  $ME^{t+1}$ ,  $MC^{t+1}$ ,  $CE^{t+1}$ , and  $MCE^{t+1}$ , which is as follows:

$$\Delta PM_{25}^{t+1} = M^{t+1} + E^{t+1} + C^{t+1} + ME^{t+1} + MC^{t+1} + CE^{t+1} + MCE^{t+1}$$
(R5)

| Markers | Equations                                                                                                                                                                           | Definitions                        |
|---------|-------------------------------------------------------------------------------------------------------------------------------------------------------------------------------------|------------------------------------|
| М       | $\frac{\partial F}{\partial x_1}x_1 + \frac{1}{2!}\frac{\partial^2 F}{\partial x_1^2}x_1^2 + \frac{1}{3!}\frac{\partial^3 F}{\partial x_1^3}x_1^3 + \cdots$                         | Pure contribution of meteorology   |
| Е       | $\frac{\partial F}{\partial x_2}x_2 + \frac{1}{2!}\frac{\partial^2 F}{\partial x_2^2}x_2^2 + \frac{1}{3!}\frac{\partial^3 F}{\partial x_2^3}x_2^3 + \cdots$                         | Pure contribution of emission      |
| С       | $\frac{\partial F}{\partial x_3}x_3 + \frac{1}{2!}\frac{\partial^2 F}{\partial x_3^2}x_3^2 + \frac{1}{3!}\frac{\partial^3 F}{\partial x_3^3}x_3^3 + \cdots$                         | Pure contribution of chemistry     |
| ME      | $\frac{1}{2!} \frac{2\partial^2 F}{\partial x_1 \partial x_2} x_1 x_2 + \frac{1}{3!} \sum_{a=1}^2 \frac{3\partial^3 F}{\partial x_1^a \partial x_2^{3-a}} x_1^a x_2^{3-a} + \cdots$ | Coupling contribution of           |
|         |                                                                                                                                                                                     | meteorology and emission           |
| CE      | $\frac{1}{2!} \frac{2\partial^2 F}{\partial x_2 \partial x_3} x_2 x_3 + \frac{1}{3!} \sum_{a=1}^2 \frac{3\partial^3 F}{\partial x_2^a \partial x_3^{3-a}} x_2^a x_3^{3-a} + \cdots$ | Coupling contribution of emission  |
|         |                                                                                                                                                                                     | and chemistry                      |
| MC      | $\frac{1}{2!} \frac{2\partial^2 F}{\partial x_1 \partial x_3} x_1 x_3 + \frac{1}{3!} \sum_{a=1}^2 \frac{3\partial^3 F}{\partial x_1^a \partial x_3^{3-a}} x_1^a x_3^{3-a} + \cdots$ | Coupling contribution of           |
|         |                                                                                                                                                                                     | meteorology and chemistry          |
| MCE     | $\frac{1}{3!} \left( \frac{\partial^3 F}{\partial x_1 \partial x_2 \partial x_3} 6 x_1 x_2 x_3 \right) + \cdots$                                                                    | Coupling contribution of emission, |
|         |                                                                                                                                                                                     | meteorology and chemistry          |

Table R1 Definition of different factors considers in the QDA method

**1.2 Algorithms of the QDA and its implementation in model**

The QDA method uses a similar algorithms to the factor separation method introduced by Stein and Alpert (1993) to calculated the terms in Eq. (R3). By setting  $x_i$  (i = 1,2,3) in Eq. (R2) to either 1 or 0, we can simply obtain following equations:

$$F(x_1, 0, 0) - F(0, 0, 0) = \frac{\partial F}{\partial x_1} x_1 + \frac{1}{2!} \frac{\partial^2 F}{\partial x_1^2} x_1^2 + \frac{1}{3!} \frac{\partial^3 F}{\partial x_1^3} x_1^3 + \dots = M$$
(R6)

$$F(0, x_2, 0) - F(0, 0, 0) = \frac{\partial F}{\partial x_2} x_2 + \frac{1}{2!} \frac{\partial^2 F}{\partial x_2^2} x_2^2 + \frac{1}{3!} \frac{\partial^3 F}{\partial x_2^3} x_2^3 + \dots = E$$
(R7)

$$F(0,0,x_3) - F(0,0,0) = \frac{\partial F}{\partial x_3} x_3 + \frac{1}{2!} \frac{\partial^2 F}{\partial x_3^2} x_3^2 + \frac{1}{3!} \frac{\partial^3 F}{\partial x_3^3} x_3^3 + \dots = C$$
(R8)

$$F(x_1, x_2, 0) - F(0, 0, 0) = \frac{\partial F}{\partial x_1} x_1 + \frac{\partial F}{\partial x_2} x_2 + \frac{1}{2!} \left( \frac{\partial^2 F}{\partial x_1^2} x_1^2 + \frac{\partial^2 F}{\partial x_2^2} x_2^2 + 2 \frac{\partial^2 F}{\partial x_1 \partial x_2} x_1 x_2 \right) + \dots = M + E + ME$$
(R9)

$$F(x_1, 0, x_3) - F(0, 0, 0) = \frac{\partial F}{\partial x_1} x_1 + \frac{\partial F}{\partial x_3} x_3 + \frac{1}{2!} \left( \frac{\partial^2 F}{\partial x_1^2} x_1^2 + \frac{\partial^2 F}{\partial x_3^2} x_3^2 + 2 \frac{\partial^2 F}{\partial x_1 \partial x_3} x_1 x_3 \right) + \dots = M + C + MC$$
(R10)

$$F(0, x_2, x_3) - F(0, 0, 0) = \frac{\partial F}{\partial x_2} x_2 + \frac{\partial F}{\partial x_3} x_3 + \frac{1}{2!} \left( \frac{\partial^2 F}{\partial x_2^2} x_2^2 + \frac{\partial^2 F}{\partial x_3^2} x_3^2 + 2 \frac{\partial^2 F}{\partial x_2 \partial x_3} x_2 x_3 \right) + \dots = E + C + CE$$
(R11)

where  $F(x_1, 0, 0)$ ,  $F(0, x_2, 0)$ ,  $F(0, 0, x_3)$  can be calculated by the simulation that only considers meteorology, emission, and chemistry process from t to t+1 step, respectively (Table R1);  $F(x_1, x_2, 0)$ ,  $F(x_1, 0, x_3)$ ,  $F(0, x_2, x_3)$ can be calculated by the simulation that does not including chemistry, emission, and meteorology process from t to t+1 step, respectively. We define these simulations as the accompanying simulation, since their concentrations were updated by the base simulation at each model step as we said in following content. According to Eq. R1 and Eq. R2, the values of F(0,0,0) and  $F(x_1, x_2, x_3)$  can be obtained from the base simulation. Based on these equations, each term in Eq. (R3) can be simply calculated by:

$$M^{t+1} = F(x_1, 0, 0)|_{PM_{2,5}^t} - F(0, 0, 0)|_{PM_{2,5}^t}$$
(R12)

$$E^{t+1} = F(0, x_2, 0)|_{PM_{2.5}^t} - F(0, 0, 0)|_{PM_{2.5}^t}$$
(R13)

$$C^{t+1} = F(0,0,x_3)|_{PM_{2.5}^t} - F(0,0,0)|_{PM_{2.5}^t}$$
(R14)

$$ME^{t+1} = F(x_1, x_2, 0)|_{PM_{2.5}^t} - F(x_1, 0, 0)|_{PM_{2.5}^t} - F(0, x_2, 0)|_{PM_{2.5}^t} + F(0, 0, 0)|_{PM_{2.5}^t}$$
(R15)

$$MC^{t+1} = F(x_1, 0, x_3)|_{PM_{2.5}^t} - F(x_1, 0, 0)|_{PM_{2.5}^t} - F(0, 0, x_3)|_{PM_{2.5}^t} + F(0, 0, 0)|_{PM_{2.5}^t}$$
(R16)

$$CE^{t+1} = F(0, x_2, x_3)|_{PM_{2.5}^t} - F(0, x_2, 0)|_{PM_{2.5}^t} - F(0, 0, x_3)|_{PM_{2.5}^t} + F(0, 0, 0)|_{PM_{2.5}^t}$$
(R17)

$$MCE^{t+1} = F(x_1, x_2, x_3)|_{PM_{2.5}^t} + \left(F(x_1, 0, 0)|_{PM_{2.5}^t} + F(0, x_2, 0)|_{PM_{2.5}^t} + F(0, 0, x_3)|_{PM_{2.5}^t}\right) - \left(F(x_1, x_2, 0)|_{PM_{2.5}^t} + F(x_1, 0, x_3)|_{PM_{2.5}^t} + F(0, x_2, x_3)|_{PM_{2.5}^t}\right) - F(0, 0, 0)|_{PM_{2.5}^t}$$
(R18)

where  $F|_{PM_{2.5}^t}$  denote the simulated PM2.5 concentration with  $PM_{2.5}^t$  as the initial condition. Based on Eq. (R1) and Eq. (R2), the values of  $F(0,0,0)|_{PM_{2.5}^t}$  and  $F(x_1,x_2,x_3)|_{PM_{2.5}^t}$  can be simply obtained from the base simulation, while the other six values are obtained from the results of six accompanying simulations. Since the accompanying simulations at each time step use  $PM_{2.5}^t$  as initial condition, the concentrations of PM2.5 and other species in the accompanying simulation will be updated by the base simulation at the start of each model step. For example, the simulation  $F(0,0,x_3)|_{PM_{2.5}^t}$  is run from t to t+1 step without including meteorology processes and emissions anywhere in the modeling, then a new PM2.5 concentration will be obtained from the base simulation for the next time step to drive  $F(0,0,x_3)|_{PM_{2.5}^{t+1}}$ .

Therefore, in  $F(0,0,x_3)$ , the meteorology processes and emissions are absent for the entire simulation but the concentrations of PM2.5 and other species in each model grid is updated by base simulation in each time step. This enables us to isolate the chemistry and emission and evaluate the contributions of different processes to the variation of PM2.5 within a time step. To achieve this, the codes of accompanying simulation were embedded in the code of base simulation so that the simulated results of each accompanying simulation at each time step can be easily and quickly updated.

|                  | Simulation
name | Processes included in the simulations | Target values       |
|------------------|--------------------|---------------------------------------|---------------------|
| Pass simulation  | S                  | All physicochemical processes         | $F(x_1, x_2, x_3),$ |
| Dase sintulation |                    |                                       | F(0,0,0)            |

Table R2. the descriptions of accompanying simulation in QDA method

|              | S 1 | Only meteorological process           | $F(x_1, 0, 0)$   |  |
|--------------|------------|---------------------------------------|------------------|--|
|              | S 2 | Only emission processes               | $F(0, x_2, 0)$   |  |
| Accompanying | S 3 | Only chemical process                 | $F(0,0,x_3)$     |  |
| simulations  | S13        | Meteorological and chemical processes | $F(x_1, 0, x_3)$ |  |
|              | S23        | Emission and chemical processes       | $F(0,x_2,x_3)$   |  |
|              | S12        | Emission and meteorological processes | $F(x_1, x_2, 0)$ |  |
|              |            |                                       |                  |  |

**1.3 Relationship with SAA (Factor Separation) and IPR method**

The scenario analysis approach (SAA) as well as its updated algorithm, Factor Separation method introduced by Stein and Alpert (1993), is an effective tool for performing model sensitivity analysis and for identifying key factors that contribute significantly to model output. Compared with the SAA method, the Factor Separation method is superior in dealing with the nonlinear process that involves two or more factors. By performing multiple sensitivity experiments with different combination of factors, the Factor Separation method allows to assess the impact of a single factor in a nonlinear system as well as the interaction between that factor and others. The similarity between the Factor Separation method and the QDA method is that they employ same algorithms to separate the contributions of different factors, while the biggest difference between the Factor Separation method and the QDA method is in the object that they resolved. As seen in Fig.R1, the Factor Separation method is designed to resolve the effects of different factors on the differences between model results from two scenarios (i.e., control simulation - base simulation). This makes the contributions of different factors resolved by Factor Separation are in a relative sense, which are dependent on the choose of scenarios. Different from the Factor Separation method, the QDA method aims to track the contributions of different factors on the variations of model results in different time steps (Fig. R1). Therefore, the results of QDA methods are in an absolute sense, which only depends on the cases we choose. In addition, in the Factor Separation method, the sensitivity experiments were run independently with the base simulation, while in the QDA method the sensitivity experiments, i.e., accompanying simulations, are coupled with the base simulation as we illustrate in Sect.1.2.

By analyzing the contribution of each process in the model, the IPR method can be used to resolved the contribution of different physical and chemical processes to the change of pollutant concentration. Considering that the emission process, chemical process and meteorological process are calculated in order in the CTMs, the IPR method is in fact equivalent to one realization of SAA method which calculated the effects of emission, meteorology and chemistry on the variation of model results by conducting three sensitivity simulations (Fig.R1). This makes the IPR method unable to consider the nonlinear effects between different factors, and results in the non-uniqueness of the results of IPR method since we can design different combinations of scenario experiments to calculate the contribution of different factors in the model process. Therefore, although both the IPR method and the QDA method aim to resolve the contribution of

different factors to the variation of model results, the QDA method is superior in handling the nonlinearity among different factors through the conduction of more sensitivity simulation.

In all, the QDA method could be seen as a combination of the Factor Separation method and IPR method. It uses the idea of Factor Separation to do the IPR analysis, which for the first time resolve the contributions of different factors as well as their interactions to the variation of model results.

**1.4 Combination with the IPR method**

Since the QDA results only gives the gross effects of emission, meteorology and chemistry processes on the variation of model results, the QDA method is combined with the IPR method to calculate the IPR results for different factors. This is achieved by applying the IPR method to each accompanying simulation. Then the results of different accompanying simulation can be decomposed as follows:

$$F(x_{1},0,0)|_{PM_{2.5}^{t}} = emit_{x1}^{t+1} + advhor_{x1}^{t+1} + advvert_{x1}^{t+1} + difhor_{x1}^{t+1} + difvert_{x1}^{t+1} + wetdep_{x1}^{t+1} + drydep_{x1}^{t+1} + gaschem_{x1}^{t+1} + ISORR_{x1}^{t+1} + SOA_{x1}^{t+1} + F(0,0,0)|_{PM_{2.5}^{t}}$$
(R19)

$$F(0, x_{2}, 0)|_{PM_{2,5}^{t}} = emit_{x_{2}}^{t+1} + advhor_{x_{2}}^{t+1} + advvert_{x_{2}}^{t+1} + difhor_{x_{2}}^{t+1} + difvert_{x_{2}}^{t+1} + wetdep_{x_{2}}^{t+1} + drydep_{x_{2}}^{t+1} + gaschem_{x_{2}}^{t+1} + ISORR_{x_{2}}^{t+1} + SOA_{x_{2}}^{t+1} + F(0,0,0)|_{PM_{2,5}^{t}}$$
(R20)

$$F(0,0,x_{3})|_{PM_{2,5}^{t}} = emit_{x_{3}}^{t+1} + advhor_{x_{3}}^{t+1} + advvert_{x_{3}}^{t+1} + difhor_{x_{3}}^{t+1} + difvert_{x_{3}}^{t+1} + wetdep_{x_{3}}^{t+1} + drydep_{x_{3}}^{t+1} + gaschem_{x_{3}}^{t+1} + ISORR_{x_{3}}^{t+1} + SOA_{x_{3}}^{t+1} + F(0,0,0)|_{PM_{2,5}^{t}}$$
(R21)

$$F(x_{1}, x_{2}, 0)|_{PM_{2,5}^{t}} = emit_{x_{1}, x_{2}}^{t+1} + advhor_{x_{1}, x_{2}}^{t+1} + advvert_{x_{1}, x_{2}}^{t+1} + difhor_{x_{1}, x_{2}}^{t+1} + difvert_{x_{1}, x_{2}}^{t+1} + wetdep_{x_{1}, x_{2}}^{t+1} + difvert_{x_{1}, x_{2}^{t+1} + difvert_{x_{1}, x_{2}}^{t+1} + difvert_{x_{1}, x_{2}}^{t+1} + difvert_{x_{1}, x_{2}}^{t+1} +$$

$$F(x_{1}, 0, x_{3})|_{PM_{2.5}^{t}} = emit_{x_{1}, x_{3}}^{t+1} + advhor_{x_{1}, x_{3}}^{t+1} + advvert_{x_{1}, x_{3}}^{t+1} + difhor_{x_{1}, x_{3}}^{t+1} + difvert_{x_{1}, x_{3}}^{t+1} + wetdep_{x_{1}, x_{3}}^{t+1} + difvert_{x_{1}, x_{3}}^{t+1} + wetdep_{x_{1}, x_{3}}^{t+1} + difvert_{x_{1}, x_{3}}^{t+1} +$$

$$F(0, x_{2}, x_{3})|_{PM_{2,5}^{t}} = emit_{x_{2}, x_{3}}^{t+1} + advhor_{x_{2}, x_{3}}^{t+1} + advvert_{x_{2}, x_{3}}^{t+1} + difhor_{x_{2}, x_{3}}^{t+1} + difvert_{x_{2}, x_{3}}^{t+1} + wetdep_{x_{2}, x_{3}}^{t+1} + difvert_{x_{2}, x_{3}}^{t+1}$$

$$F(x_{1}, x_{2}, x_{3})|_{PM_{2,5}^{t}} = emit_{x_{1}, x_{2}, x_{3}}^{t+1} + advhor_{x_{1}, x_{2}, x_{3}}^{t+1} + advvert_{x_{1}, x_{2}, x_{3}}^{t+1} + difhor_{x_{1}, x_{2}, x_{3}}^{t+1} + difvert_{x_{1}, x_{2}, x_{3}}^{t$$

where  $emit_{x_1}^{t+1}$ ,  $emit_{x_2}^{t+1}$ ,  $\cdots$ ,  $emit_{x_1,x_2,x_3}^{t+1}$  represent the IPR results for emission processes in different accompanying simulation from t to t+1 step, and so do the other processes. Note that some processes in specific accompanying simulation is equal to zero, for example the  $emit_{x_1}^{t+1}$ ,  $gaschem_{x_1}^{t+1}$ ,  $ISORR_{x_1}^{t+1}$  and  $SOA_{x_1}^{t+1}$  term in  $F(x_1, 0, 0)|_{PM_{2.5}^t}$  since  $F(x_1, 0, 0)$  only considers the meteorological processes.

Based on Eq (R19-R25), IPR results for each factor can be calculated as the same way of the contribution of each

(R26)

factor. For example, the formulation of  $M^{t+1}$  can be rewritten as follows based on IPR:  $M^{t+1} = emit_{x_2}^{t+1} + advhor_{x_2}^{t+1} + advvert_{x_2}^{t+1} + difhor_{x_2}^{t+1} + difvert_{x_2}^{t+1} + wetdep_{x_2}^{t+1} + drydep_{x_2}^{t+1} + gaschem_{x_2}^{t+1} + ISORR_{x_2}^{t+1} + SOA_{x_2}^{t+1}$

Using the same manner, the IPR results for other factors can be calculated according to Eq (R12-R25).

**QDA method:**

**Figure R1. Comparisons of QDA method with the Factor Separation method and IPR method**

**Comment 2:** Results and discussion of QDA. Because the description of QDA is insufficient, I also cannot follow the result and discussion section. Why E term showed same values through analyzed stages? Is this because emissions did not consider temporal variation through analyzed episode? The meteorological field are shown in Fig. 4, but how about the precipitation? Because the term of "wetdep" was 0.00 through stages, I felt that there was no rain. Although this was the severe haze event, without the wet deposition analysis, this episode seems to be not interesting as test case to show the QDA result. As evaluated using NOR and SOR, I like the idea to consider the formation process from the viewpoint of each specie. The result of QDA is now discussed for PM2.5; however, each specie have been evolved as different E and C terms. I would like to strongly recommend to show the same kind of analysis of Figs. 7-9 for each specie. This analysis will offer the insight into C roles on chemical formation during haze episode.

**Reply:** Thanks for your comments on our article. According to your comment, we have divided the whole comment into three small comments with detailed point-by-point responses listed below:

**(1) Why E term showed same values through analyzed stages? Is this because emissions did not consider temporal variation through analyzed episode?**

Reply: Yes, we did not consider the temporal variation of the emission throughout the analyzed episode since the

bottom-up emissions are only available at monthly resolution and it is difficult to accurately estimate the time-variation of emissions from different sectors. Also, in the QDA method, the values of E just represent the pure contribution of emission, which only considers the effects of direct emission at local space. That's why the values of E kept the same throughout the analyzed episode. To account for the effects of the temporal variation of emission on the QDA method, we re-performed the QDA analysis with the considerations of diurnal variation of emissions from different sectors. Figure R2 shows the diurnal profile of the emissions from different sectors obtained from the MIX inventory(Li et al., 2017), which generally shows higher emissions during the daytime than the nighttime. The transport and residential emission also show a double-peak pattern in their diurnal profile.

---

## Author Comment (AC3)

**Response to Referee #2 (gmd-2021-259)**

We Thank Reviewer for his/her constructive comments.

Responses to the comments:

The manuscript 'A quantitative decoupling analysis (QDA v1.0) method for the assessment of meteorological, emission and chemical contributions to fine particulate pollution' written by Junhua Wang presented the QDA method as novel way to evaluate meteorology, emission, and chemistry processes involved for the aerosol formation. Although the concept of this method is interesting, I cannot fully understand the description of method itself and therefore go through to result and discussion section well. At the current presentation quality, this manuscript cannot be considered for publication. At this round, I would like to reject this manuscript. Before considering the possible publication, I sincerely request the fundamental amendments. I wish the following major and minor comments will help to revise this manuscript.

**Reply:** The authors appreciate the reviewer for his/her constructive and up-to-point comments. We have carefully considered the comments and revised the manuscript accordingly. Please refer to our responses for more details given below.

**Major comments:**

**Comment 1:** The description of QDA and its relation to IPR. The newly developed QDA method is just the using of six accompanying simulations to calculate M, E, and C terms. In this sense, for example, to drive M term, this method seems to be identical to the SAA as described in the introduction. Actually, how to conduct six accompanying simulations is unclear. Under each time-step simulation, how can do the base-model derive each process? The detailed description of M2-M7 is required to understand the QDA method. In addition, without E term, C term cannot be driven due to the absence of precursors. Therefore, I guess that EC term inherently connected, and it could be hard to be divided. Moreover, on P4, L118, it was stated that "The above QDA method can also be combine with the IPR method to resolve more detailed information…". This statement is confusing to me because this impressed that QDA is just the using of IPR. Under the current presentation quality, it is difficult to understand QDA method and I cannot recognize this method as novel way in modeling analysis.

**Reply:** We feel sorry that we did not provide enough description on the QDA method. We have made a more detailed description on the QDA method, including its theoretical basis, algorithms, its realization in model as well as its relationship with the SAA (Factor Separation method) and IPR method, to facilitate the understanding of the QDA method and highlight the novelty of QDA method.

**1.1 Theoretical basis of the QDA method**

The QDA method is developed based on the Taylor series expansion. Considering that the $PM_{2.5}$ concentration at t step is $PM_{2.5}^t$ and the $PM_{2.5}$ concentration at t+1 step after undergoing the emission, meteorology and chemistry processes with $PM_{2.5}^t$ as initial condition is $PM_{2.5}^{t+1}$, then we could define a function F that denotes the simulated $PM_{2.5}$

concentrations with or without different processes using $PM_{2.5}^{t}$ as initial concentration, such that:

$$F(0,0,0) = PM_{2.5}^{t} \tag{R1}$$

$$F(x_1, x_2, x_3) = PM_{2.5}^{t+1} \tag{R2}$$

where $F(x_1, x_2, x_3)$ represents the simulated PM$_{2.5}$ concentration with meteorology $(x_1)$, emission $(x_2)$, and chemistry processes $(x_3)$; $F(0,0,0)$ represents the simulated PM$_{2.5}$ concentration without emission, meteorology, and chemistry processes. Therefore, the $PM_{2.5}^{t}$ and the $PM_{2.5}^{t+1}$ can be seen as the different values of function F with different input data, and the variation of PM$_{2.5}$ concentration between two timesteps can be written as:

$$\Delta PM_{2.5}^{t+1} = PM_{2.5}^{t+1} - PM_{2.5}^{t} = F(x_1, x_2, x_3) - F(0,0,0) \tag{R3}$$

where $\Delta PM_{2.5}^{t+1}$ represents the variation of PM$_{2.5}$ concentration from t to t+1 step,

According to Taylor series expansion, the function $F$ can be decomposed as follows:

$$F(x_1, x_2, x_3) - F(0,0,0) = \sum_{i=1}^{3} \frac{\partial F}{\partial x_i} x_i + \frac{1}{2!}\left(\sum_{i=1}^{3} \frac{\partial^2 F}{\partial x_i^2} x_i^2 + 2\frac{\partial^2 F}{\partial x_1 \partial x_2} x_1 x_2 + 2\frac{\partial^2 F}{\partial x_2 \partial x_3} x_2 x_3 + 2\frac{\partial^2 F}{\partial x_1 \partial x_3} x_1 x_3\right) +$$

$$\frac{1}{3!}\left(\sum_{i=1}^{3} \frac{\partial^3 F}{\partial x_i^3} x_i^3 + \sum_{a=1}^{2} 3\frac{\partial^3 F}{\partial x_1^a \partial x_2^{3-a}} x_1^a x_2^{3-a} + \sum_{a=1}^{2} 3\frac{\partial^3 F}{\partial x_2^a \partial x_3^{3-a}} x_2^a x_3^{3-a} + \sum_{a=1}^{2} 3\frac{\partial^3 F}{\partial x_1^a \partial x_3^{3-a}} x_1^a x_3^{3-a} +$$

$$\frac{\partial^3 F}{\partial x_1 \partial x_2 \partial x_3} 6 x_1 x_2 x_3\right) + \cdots + o^n \tag{R4}$$

Based on this equation, the terms that only containing a single partial derivative to $x_1$, $x_2$, and $x_3$ (including any higher-order derivatives) are defined as pure contribution of the meteorology (M), emission (E) and chemistry (C) processes to the variation of PM$_{2.5}$ concentrations. Therefore, the term $\frac{\partial F}{\partial x_1} x_1 + \frac{1}{2!}\frac{\partial^2 F}{\partial x_1^2} x_1^2 + \frac{1}{3!}\frac{\partial^3 F}{\partial x_1^3} x_1^3 + \cdots$ in Eq.(R2) is defined as E, term $\frac{\partial F}{\partial x_2} x_2 + \frac{1}{2!}\frac{\partial^2 F}{\partial x_2^2} x_2^2 + \frac{1}{3!}\frac{\partial^3 F}{\partial x_2^3} x_2^3 + \cdots$ is defined as M, and the term $\frac{\partial F}{\partial x_3} x_3 + \frac{1}{2!}\frac{\partial^2 F}{\partial x_3^2} x_3^2 + \frac{1}{3!}\frac{\partial^3 F}{\partial x_3^3} x_3^3 + \cdots$ is defined as C. The cross terms then represent the interaction among different drivers, for example the term $\frac{1}{2!}\frac{2\partial^2 F}{\partial x_1 \partial x_2} x_1 x_2 +$ $\frac{1}{3!}\sum_{a=1}^{2} \frac{3\partial^3 F}{\partial x_1^a \partial x_2^{3-a}} x_1^a x_2^{3-a} + \cdots$ is defined as the interactions between meteorology and emission (ME), the term $\frac{1}{3!}\left(\frac{\partial^3 F}{\partial x_1 \partial x_2 \partial x_3} 6 x_1 x_2 x_3\right) + \cdots$ is defined as the interactions among emission, meteorology and chemistry processes (MCE). Detailed definitions to the different factors we resolved are available in Table R1. Note that pure here as well as elsewhere in the paper, is in the relative sense meaning that the effect due to one factor is separated from the other chosen factors. For example, the pure contribution of emission is only due to the direct emission at local space. The variation of PM$_{2.5}$ concentration after emission process are seen as the contribution of meteorology and chemistry. Therefore, the values of E in the QDA method cannot represent the whole effects of emission in the common sense.

According to these definitions, the PM$_{2.5}$ variations from t to t+1 step can be written as the sum of $M^{t+1}$, $E^{t+1}$, $C^{t+1}$, $ME^{t+1}$, $MC^{t+1}$, $CE^{t+1}$, and $MCE^{t+1}$, which is as follows:

$$\Delta PM_{2.5}^{t+1} = M^{t+1} + E^{t+1} + C^{t+1} + ME^{t+1} + MC^{t+1} + CE^{t+1} + MCE^{t+1} \tag{R5}$$

**Table R1 Definition of different factors considers in the QDA method**

| Markers | Equations | Definitions |
|---|---|---|
| M | $\dfrac{\partial F}{\partial x_1}x_1 + \dfrac{1}{2!}\dfrac{\partial^2 F}{\partial x_1^2}x_1^2 + \dfrac{1}{3!}\dfrac{\partial^3 F}{\partial x_1^3}x_1^3 + \cdots$ | Pure contribution of meteorology |
| E | $\dfrac{\partial F}{\partial x_2}x_2 + \dfrac{1}{2!}\dfrac{\partial^2 F}{\partial x_2^2}x_2^2 + \dfrac{1}{3!}\dfrac{\partial^3 F}{\partial x_2^3}x_2^3 + \cdots$ | Pure contribution of emission |
| C | $\dfrac{\partial F}{\partial x_3}x_3 + \dfrac{1}{2!}\dfrac{\partial^2 F}{\partial x_3^2}x_3^2 + \dfrac{1}{3!}\dfrac{\partial^3 F}{\partial x_3^3}x_3^3 + \cdots$ | Pure contribution of chemistry |
| ME | $\dfrac{1}{2!}\dfrac{2\partial^2 F}{\partial x_1 \partial x_2}x_1 x_2 + \dfrac{1}{3!}\displaystyle\sum_{a=1}^{2}\dfrac{3\partial^3 F}{\partial x_1^a \partial x_2^{3-a}}x_1^a x_2^{3-a} + \cdots$ | Coupling contribution of meteorology and emission |
| CE | $\dfrac{1}{2!}\dfrac{2\partial^2 F}{\partial x_2 \partial x_3}x_2 x_3 + \dfrac{1}{3!}\displaystyle\sum_{a=1}^{2}\dfrac{3\partial^3 F}{\partial x_2^a \partial x_3^{3-a}}x_2^a x_3^{3-a} + \cdots$ | Coupling contribution of emission and chemistry |
| MC | $\dfrac{1}{2!}\dfrac{2\partial^2 F}{\partial x_1 \partial x_3}x_1 x_3 + \dfrac{1}{3!}\displaystyle\sum_{a=1}^{2}\dfrac{3\partial^3 F}{\partial x_1^a \partial x_3^{3-a}}x_1^a x_3^{3-a} + \cdots$ | Coupling contribution of meteorology and chemistry |
| MCE | $\dfrac{1}{3!}\left(\dfrac{\partial^3 F}{\partial x_1 \partial x_2 \partial x_3}6 x_1 x_2 x_3\right) + \cdots$ | Coupling contribution of emission, meteorology and chemistry |

**1.2 Algorithms of the QDA and its implementation in model**

The QDA method uses a similar algorithms to the factor separation method introduced by Stein and Alpert (1993) to calculated the terms in Eq. (R3). By setting $x_i$ $(i = 1,2,3)$ in Eq. (R2) to either 1 or 0, we can simply obtain following equations:

$$F(x_1,0,0) - F(0,0,0) = \frac{\partial F}{\partial x_1}x_1 + \frac{1}{2!}\frac{\partial^2 F}{\partial x_1^2}x_1^2 + \frac{1}{3!}\frac{\partial^3 F}{\partial x_1^3}x_1^3 + \cdots = M \tag{R6}$$

$$F(0,x_2,0) - F(0,0,0) = \frac{\partial F}{\partial x_2}x_2 + \frac{1}{2!}\frac{\partial^2 F}{\partial x_2^2}x_2^2 + \frac{1}{3!}\frac{\partial^3 F}{\partial x_2^3}x_2^3 + \cdots = E \tag{R7}$$

$$F(0,0,x_3) - F(0,0,0) = \frac{\partial F}{\partial x_3}x_3 + \frac{1}{2!}\frac{\partial^2 F}{\partial x_3^2}x_3^2 + \frac{1}{3!}\frac{\partial^3 F}{\partial x_3^3}x_3^3 + \cdots = C \tag{R8}$$

$$F(x_1,x_2,0) - F(0,0,0) = \frac{\partial F}{\partial x_1}x_1 + \frac{\partial F}{\partial x_2}x_2 + \frac{1}{2!}\left(\frac{\partial^2 F}{\partial x_1^2}x_1^2 + \frac{\partial^2 F}{\partial x_2^2}x_2^2 + 2\frac{\partial^2 F}{\partial x_1 \partial x_2}x_1 x_2\right) + \cdots = M + E + ME \tag{R9}$$

$$F(x_1,0,x_3) - F(0,0,0) = \frac{\partial F}{\partial x_1}x_1 + \frac{\partial F}{\partial x_3}x_3 + \frac{1}{2!}\left(\frac{\partial^2 F}{\partial x_1^2}x_1^2 + \frac{\partial^2 F}{\partial x_3^2}x_3^2 + 2\frac{\partial^2 F}{\partial x_1 \partial x_3}x_1 x_3\right) + \cdots = M + C + MC \tag{R10}$$

$$F(0,x_2,x_3) - F(0,0,0) = \frac{\partial F}{\partial x_2}x_2 + \frac{\partial F}{\partial x_3}x_3 + \frac{1}{2!}\left(\frac{\partial^2 F}{\partial x_2^2}x_2^2 + \frac{\partial^2 F}{\partial x_3^2}x_3^2 + 2\frac{\partial^2 F}{\partial x_2 \partial x_3}x_2 x_3\right) + \cdots = E + C + CE \tag{R11}$$

where $F(x_1,0,0)$, $F(0,x_2,0)$, $F(0,0,x_3)$ can be calculated by the simulation that only considers meteorology, emission, and chemistry process from t to t+1 step, respectively (Table R1); $F(x_1,x_2,0)$, $F(x_1,0,x_3)$, $F(0,x_2,x_3)$ can be calculated by the simulation that does not including chemistry, emission, and meteorology process from t to t+1 step, respectively. We define these simulations as the accompanying simulation, since their concentrations were updated by the base simulation at each model step as we said in following content. According to Eq. R1 and Eq. R2, the values of $F(0,0,0)$ and $F(x_1,x_2,x_3)$ can be obtained from the base simulation. Based on these equations, each term in Eq.

(R3) can be simply calculated by:

$$M^{t+1} = F(x_1,0,0)|_{PM_{2.5}^t} - F(0,0,0)|_{PM_{2.5}^t} \tag{R12}$$

$$E^{t+1} = F(0,x_2,0)|_{PM_{2.5}^t} - F(0,0,0)|_{PM_{2.5}^t} \tag{R13}$$

$$C^{t+1} = F(0,0,x_3)|_{PM_{2.5}^t} - F(0,0,0)|_{PM_{2.5}^t} \tag{R14}$$

$$ME^{t+1} = F(x_1,x_2,0)|_{PM_{2.5}^t} - F(x_1,0,0)|_{PM_{2.5}^t} - F(0,x_2,0)|_{PM_{2.5}^t} + F(0,0,0)|_{PM_{2.5}^t} \tag{R15}$$

$$MC^{t+1} = F(x_1,0,x_3)|_{PM_{2.5}^t} - F(x_1,0,0)|_{PM_{2.5}^t} - F(0,0,x_3)|_{PM_{2.5}^t} + F(0,0,0)|_{PM_{2.5}^t} \tag{R16}$$

$$CE^{t+1} = F(0,x_2,x_3)|_{PM_{2.5}^t} - F(0,x_2,0)|_{PM_{2.5}^t} - F(0,0,x_3)|_{PM_{2.5}^t} + F(0,0,0)|_{PM_{2.5}^t} \tag{R17}$$

$$MCE^{t+1} = F(x_1,x_2,x_3)|_{PM_{2.5}^t} + \left(F(x_1,0,0)|_{PM_{2.5}^t} + F(0,x_2,0)|_{PM_{2.5}^t} + F(0,0,x_3)|_{PM_{2.5}^t}\right) - \left(F(x_1,x_2,0)|_{PM_{2.5}^t} + \right.$$

$$\left. F(x_1,0,x_3)|_{PM_{2.5}^t} + F(0,x_2,x_3)|_{PM_{2.5}^t}\right) - F(0,0,0)|_{PM_{2.5}^t} \tag{R18}$$

where $F|_{PM_{2.5}^t}$ denote the simulated PM$_{2.5}$ concentration with $PM_{2.5}^t$ as the initial condition. Based on Eq. (R1) and Eq. (R2), the values of $F(0,0,0)|_{PM_{2.5}^t}$ and $F(x_1,x_2,x_3)|_{PM_{2.5}^t}$ can be simply obtained from the base simulation, while the other six values are obtained from the results of six accompanying simulations. Since the accompanying simulations at each time step use $PM_{2.5}^t$ as initial condition, the concentrations of PM$_{2.5}$ and other species in the accompanying simulation will be updated by the base simulation at the start of each model step. For example, the simulation $F(0,0,x_3)|_{PM_{2.5}^t}$ is run from t to t+1 step without including meteorology processes and emissions anywhere in the modeling, then a new PM$_{2.5}$ concentration will be obtained from the base simulation for the next time step to drive $F(0,0,x_3)|_{PM_{2.5}^{t+1}}$.

Therefore, in $F(0,0,x_3)$, the meteorology processes and emissions are absent for the entire simulation but the concentrations of PM$_{2.5}$ and other species in each model grid is updated by base simulation in each time step. This enables us to isolate the chemistry and emission and evaluate the contributions of different processes to the variation of PM$_{2.5}$ within a time step. To achieve this, the codes of accompanying simulation were embedded in the code of base simulation so that the simulated results of each accompanying simulation at each time step can be easily and quickly updated.

**Table R2. the descriptions of accompanying simulation in QDA method**

| | Simulation name | Processes included in the simulations | Target values |
|---|---|---|---|
| Base simulation | S | All physicochemical processes | $F(x_1,x_2,x_3)$, $F(0,0,0)$ |

| | S1 | Only meteorological process | $F(x_1, 0, 0)$ |
|---|---|---|---|
| | S2 | Only emission processes | $F(0, x_2, 0)$ |
| Accompanying simulations | S3 | Only chemical process | $F(0, 0, x_3)$ |
| | S13 | Meteorological and chemical processes | $F(x_1, 0, x_3)$ |
| | S23 | Emission and chemical processes | $F(0, x_2, x_3)$ |
| | S12 | Emission and meteorological processes | $F(x_1, x_2, 0)$ |

**1.3 Relationship with SAA (Factor Separation) and IPR method**

The scenario analysis approach (SAA) as well as its updated algorithm, Factor Separation method introduced by Stein and Alpert (1993), is an effective tool for performing model sensitivity analysis and for identifying key factors that contribute significantly to model output. Compared with the SAA method, the Factor Separation method is superior in dealing with the nonlinear process that involves two or more factors. By performing multiple sensitivity experiments with different combination of factors, the Factor Separation method allows to assess the impact of a single factor in a nonlinear system as well as the interaction between that factor and others. The similarity between the Factor Separation method and the QDA method is that they employ same algorithms to separate the contributions of different factors, while the biggest difference between the Factor Separation method and the QDA method is in the object that they resolved. As seen in Fig.R1, the Factor Separation method is designed to resolve the effects of different factors on the differences between model results from two scenarios (i.e., control simulation – base simulation). This makes the contributions of different factors resolved by Factor Separation are in a relative sense, which are dependent on the choose of scenarios. Different from the Factor Separation method, the QDA method aims to track the contributions of different factors on the variations of model results in different time steps (Fig. R1). Therefore, the results of QDA methods are in an absolute sense, which only depends on the cases we choose. In addition, in the Factor Separation method, the sensitivity experiments were run independently with the base simulation, while in the QDA method the sensitivity experiments, i.e., accompanying simulations, are coupled with the base simulation as we illustrate in Sect.1.2.

By analyzing the contribution of each process in the model, the IPR method can be used to resolved the contribution of different physical and chemical processes to the change of pollutant concentration. Considering that the emission process, chemical process and meteorological process are calculated in order in the CTMs, the IPR method is in fact equivalent to one realization of SAA method which calculated the effects of emission, meteorology and chemistry on the variation of model results by conducting three sensitivity simulations (Fig.R1). This makes the IPR method unable to consider the nonlinear effects between different factors, and results in the non-uniqueness of the results of IPR method since we can design different combinations of scenario experiments to calculate the contribution of different factors in the model process. Therefore, although both the IPR method and the QDA method aim to resolve the contribution of

different factors to the variation of model results, the QDA method is superior in handling the nonlinearity among different factors through the conduction of more sensitivity simulation.

In all, the QDA method could be seen as a combination of the Factor Separation method and IPR method. It uses the idea of Factor Separation to do the IPR analysis, which for the first time resolve the contributions of different factors as well as their interactions to the variation of model results.

**1.4 Combination with the IPR method**

Since the QDA results only gives the gross effects of emission, meteorology and chemistry processes on the variation of model results, the QDA method is combined with the IPR method to calculate the IPR results for different factors. This is achieved by applying the IPR method to each accompanying simulation. Then the results of different accompanying simulation can be decomposed as follows:

$$F(x_1,0,0)|_{PM_{2.5}^t} = emit_{x1}^{t+1} + advhor_{x1}^{t+1} + advvert_{x1}^{t+1} + difhor_{x1}^{t+1} + difvert_{x1}^{t+1} + wetdep_{x1}^{t+1} + drydep_{x1}^{t+1} +$$

$$gaschem_{x1}^{t+1} + ISORR_{x1}^{t+1} + SOA_{x1}^{t+1} + F(0,0,0)|_{PM_{2.5}^t} \tag{R19}$$

$$F(0,x_2,0)|_{PM_{2.5}^t} = emit_{x_2}^{t+1} + advhor_{x_2}^{t+1} + advvert_{x_2}^{t+1} + difhor_{x_2}^{t+1} + difvert_{x_2}^{t+1} + wetdep_{x_2}^{t+1} + drydep_{x_2}^{t+1} +$$

$$gaschem_{x_2}^{t+1} + ISORR_{x_2}^{t+1} + SOA_{x_2}^{t+1} + F(0,0,0)|_{PM_{2.5}^t} \tag{R20}$$

$$F(0,0,x_3)|_{PM_{2.5}^t} = emit_{x_3}^{t+1} + advhor_{x_3}^{t+1} + advvert_{x_3}^{t+1} + difhor_{x_3}^{t+1} + difvert_{x_3}^{t+1} + wetdep_{x_3}^{t+1} + drydep_{x_3}^{t+1} +$$

$$gaschem_{x_3}^{t+1} + ISORR_{x_3}^{t+1} + SOA_{x_3}^{t+1} + F(0,0,0)|_{PM_{2.5}^t} \tag{R21}$$

$$F(x_1,x_2,0)|_{PM_{2.5}^t} = emit_{x_1,x_2}^{t+1} + advhor_{x_1,x_2}^{t+1} + advvert_{x_1,x_2}^{t+1} + difhor_{x_1,x_2}^{t+1} + difvert_{x_1,x_2}^{t+1} + wetdep_{x_1,x_2}^{t+1} +$$

$$drydep_{x_1,x_2}^{t+1} + gaschem_{x_1,x_2}^{t+1} + ISORR_{x_1,x_2}^{t+1} + SOA_{x_1,x_2}^{t+1} + F(0,0,0)|_{PM_{2.5}^t} \tag{R22}$$

$$F(x_1,0,x_3)|_{PM_{2.5}^t} = emit_{x_1,x_3}^{t+1} + advhor_{x_1,x_3}^{t+1} + advvert_{x_1,x_3}^{t+1} + difhor_{x_1,x_3}^{t+1} + difvert_{x_1,x_3}^{t+1} + wetdep_{x_1,x_3}^{t+1} +$$

$$drydep_{x_1,x_3}^{t+1} + gaschem_{x_1,x_3}^{t+1} + ISORR_{x_1,x_3}^{t+1} + SOA_{x_1,x_3}^{t+1} + F(0,0,0)|_{PM_{2.5}^t} \tag{R23}$$

$$F(0,x_2,x_3)|_{PM_{2.5}^t} = emit_{x_2,x_3}^{t+1} + advhor_{x_2,x_3}^{t+1} + advvert_{x_2,x_3}^{t+1} + difhor_{x_2,x_3}^{t+1} + difvert_{x_2,x_3}^{t+1} + wetdep_{x_2,x_3}^{t+1} +$$

$$drydep_{x_2,x_3}^{t+1} + gaschem_{x_2,x_3}^{t+1} + ISORR_{x_2,x_3}^{t+1} + SOA_{x_2,x_3}^{t+1} + F(0,0,0)|_{PM_{2.5}^t} \tag{R24}$$

$$F(x_1,x_2,x_3)|_{PM_{2.5}^t} = emit_{x_1,x_2,x_3}^{t+1} + advhor_{x_1,x_2,x_3}^{t+1} + advvert_{x_1,x_2,x_3}^{t+1} + difhor_{x_1,x_2,x_3}^{t+1} + difvert_{x_1,x_2,x_3}^{t+1} +$$

$$wetdep_{x_1,x_2,x_3}^{t+1} + drydep_{x_1,x_2,x_3}^{t+1} + gaschem_{x_1,x_2,x_3}^{t+1} + ISORR_{x_1,x_2,x_3}^{t+1} + SOA_{x_1,x_2,x_3}^{t+1} + F(0,0,0)|_{PM_{2.5}^t} \tag{R25}$$

where $emit_{x1}^{t+1}, emit_{x_2}^{t+1}, \cdots, emit_{x_1,x_2,x_3}^{t+1}$ represent the IPR results for emission processes in different accompanying simulation from t to t+1 step, and so do the other processes. Note that some processes in specific accompanying simulation is equal to zero, for example the $emit_{x_1}^{t+1}, gaschem_{x_1}^{t+1}, ISORR_{x_1}^{t+1}$ and $SOA_{x_1}^{t+1}$ term in $F(x_1,0,0)|_{PM_{2.5}^t}$ since $F(x_1,0,0)$ only considers the meteorological processes.

Based on Eq (R19-R25), IPR results for each factor can be calculated as the same way of the contribution of each factor. For example, the formulation of $M^{t+1}$ can be rewritten as follows based on IPR:

$$M^{t+1} = emit_{x_2}^{t+1} + advhor_{x_2}^{t+1} + advvert_{x_2}^{t+1} + difhor_{x_2}^{t+1} + difvert_{x_2}^{t+1} + wetdep_{x_2}^{t+1} + drydep_{x_2}^{t+1} + gaschem_{x_2}^{t+1} + ISORR_{x_2}^{t+1} + SOA_{x_2}^{t+1} \qquad (R26)$$

Using the same manner, the IPR results for other factors can be calculated according to Eq (R12-R25).

[Figure]

**Figure R1. Comparisons of QDA method with the Factor Separation method and IPR method**

**Comment 2:** Results and discussion of QDA. Because the description of QDA is insufficient, I also cannot follow the result and discussion section. Why E term showed same values through analyzed stages? Is this because emissions did not consider temporal variation through analyzed episode? The meteorological field are shown in Fig. 4, but how about the precipitation? Because the term of "wetdep" was 0.00 through stages, I felt that there was no rain. Although this was the severe haze event, without the wet deposition analysis, this episode seems to be not interesting as test case to show the QDA result. As evaluated using NOR and SOR, I like the idea to consider the formation process from the viewpoint of each specie. The result of QDA is now discussed for PM2.5; however, each specie have been evolved as different E and C terms. I would like to strongly recommend to show the same kind of analysis of Figs. 7-9 for each specie. This analysis will offer the insight into C roles on chemical formation during haze episode.

**Reply:** Thanks for your comments on our article. According to your comment, we have divided the whole comment into three small comments with detailed point-by-point responses listed below:

**(1) Why E term showed same values through analyzed stages? Is this because emissions did not consider temporal variation through analyzed episode?**

**Reply:** Yes, we did not consider the temporal variation of the emission throughout the analyzed episode since the

bottom-up emissions are only available at monthly resolution and it is difficult to accurately estimate the time-variation of emissions from different sectors. Also, in the QDA method, the values of E just represent the pure contribution of emission, which only considers the effects of direct emission at local space. That's why the values of E kept the same throughout the analyzed episode. To account for the effects of the temporal variation of emission on the QDA method, we re-performed the QDA analysis with the considerations of diurnal variation of emissions from different sectors. Figure R2 shows the diurnal profile of the emissions from different sectors obtained from the MIX inventory(Li et al., 2017), which generally shows higher emissions during the daytime than the nighttime. The transport and residential emission also show a double-peak pattern in their diurnal profile.

[Figure]

**Figure R2 the diurnal profile of emissions from different sectors**

Figure R3 shows the updated time series of the calculated contributions of emission, meteorology, chemistry and their interactions to the $PM_{2.5}$ variations. Compared with the QDA results without considerations of emission variation, the time-variation of the updated QDA results is generally larger. For example, the calculated meteorological contributions (M) ranges from -48.7 to 7.4 $\mu g \cdot m^{-3} \cdot h^{-1}$ when the emission variation was considered, larger than the values of M (-42–8 $\mu g \cdot m^{-3} \cdot h^{-1}$) without the consideration of emission variation. The time-varying emission also induces larger variation in the contribution of the coupling effects of emission and chemistry, with the calculated contribution of EC ranging from 0 to 1.8 $\mu g \cdot m^{-3} \cdot h^{-1}$ higher than the values of EC (0.1–1.3 $\mu g \cdot m^{-3} \cdot h^{-1}$) without consideration of emission variation.

[Figure]

**Figure R3. time series of hourly PM$_{2.5}$ variations between adjacent hours (black lines) from 17 Feb to 28 Feb, 2014 as well as the contributions of**

Figure R4 gives more detailed results for the diurnal variation of the contributions of different factors during the first three stage of episode over the Beijing area. The last stage is not analyzed since it did not last for one day. According to fig. R4a, the PM$_{2.5}$ concentration decreased by 14.3 μg m−3 during the period of Stage 1 (from 91.2 μg m−3 at 00:00 LST to 76.9 μg m−3 at 23:00 LST). But in Stage 2 (fig. R4b), the PM$_{2.5}$ concentration exhibits a significant monotonic growth, with a daily increment of 37.1 μg m$^{-3}$ (from 56.7 μg m$^{-3}$ at 00:00 LST to 93.8 μg m$^{-3}$ at 23:00 LST). The diurnal variation of PM$_{2.5}$ is small in Stage 3 (fig. R3c), only increased by about 6.3 μg m$^{-3}$. This indicates that Stage 2 has the most favorable environmental conditions for the growth of PM$_{2.5}$, leading to the most significant change of PM$_{2.5}$ concentration compared to other stages. The daily concentration changes in Stage 1 and Stage 3 are both small, indicates that the environment in these periods tend to maintain the stability of PM$_{2.5}$.

The QDA results suggest that the contribution of pure meteorology contribution (M) was generally negative during the first stage, especially at forenoon (05:00–8:00 LST) and afternoon (15:00–17:00 LST) with estimated values of M up to -3 $μg \cdot m^{-3} \cdot h^{-1}$. The scavenging effects of M almost become zero during 12:00–15:00 LST. In addition, the contribution of the interaction between meteorology and chemistry become larger, together with the larger pure contribution of emission (E) and chemistry (C), making the PM$_{2.5}$ concentration increased slightly during that time.

However, the values of M turned to be positive during most time of stage 2 especially during the nighttime (fig.R3e), with estimated values of M up to 2.2 $μg \cdot m^{-3} \cdot h^{-1}$, much higher than the values of E and C. This suggest that the meteorology dominated the increases of PM$_{2.5}$ at the nighttime of stage 2, and that the control of local emission, with the values of E only ranging from 0.3 to 0.7 $μg \cdot m^{-3} \cdot h^{-1}$ during nighttime, may only has little effects on the control of PM$_{2.5}$. However, the meteorology contribution contains the contribution of transportation of non-local PM$_{2.5}$ concentrations, thus it should be more effective to control the emissions outside Beijing during stage 2, which would effectively slow down the accumulation of PM$_{2.5}$ and may prevent the occurrence of potential heavy haze episode.

Although the pure contribution of meteorology (M) become negative during 12:00–18:00 LST, ranging from -1.6 to -0.1 $\mu g \cdot m^{-3} \cdot h^{-1}$, the coupling effects between meteorology and chemical (MC) become positive during that time (0.1–0.6 $\mu g \cdot m^{-3} \cdot h^{-1}$), which indicates that the meteorology condition favors the chemical production of PM$_{2.5}$. The pure contribution of emission and chemistry also become positive and together with MC counteract the scavenging effects of meteorology. This suggests that local emission control both for PM$_{2.5}$ and its precursors is needed if we aimed to migrate the PM$_{2.5}$ pollution at this time.

At stage 3 (fig. R4f), the concentration of PM$_{2.5}$ was maintained at a high level with small fluctuation, which indicates that the contributions of different factors generally reach an equilibrium. The pure contribution of meteorology was relatively weak during the nighttime, but indicates significant scavenging effects during 13:00–19:00 LST with the values of M ranging from -5.1 to -2.7 $\mu g \cdot m^{-3} \cdot h^{-1}$. However, the values of E and CE also increased significantly during that time especially for CE, with maximum values up to 1.2 and 1.6 $\mu g \cdot m^{-3} \cdot h^{-1}$, respectively, which counteract the negative contribution of meteorology. As a result, the PM$_{2.5}$ concentration only slightly decreased during that time. This suggests that for this case, it is still necessary to control the local emissions of PM$_{2.5}$ and its precursor at stage 3.

[Figure]

**Figure R4. Diurnal variation of the vertical average concentrations of PM$_{2.5}$ as well as its compositions (a-c), and that of the contributions of different factors (d-f) as well as different meteorological parameters (g-i) during the first three stage.**

**(2)  The meteorological field are shown in Fig. 4, but how about the precipitation? Because the term of "wetdep" was 0.00 through stages, I felt that there was no rain. Although this was the severe haze event, without the wet deposition analysis, this episode seems to be not interesting as test case to show the QDA result.**

**Reply:** Yes, there was no rain through the whole episode, so the "wetdep" term was equal to 0.00 through stages. We agree with the reviewer that the wet deposition analysis is interesting for analyzing effects the wet deposition on the

PM$_{2.5}$ concentrations, especially during the summertime when wet deposition may exert significant impacts on removal of air pollutants. However, the PM$_{2.5}$ pollution over Beijing-Tianjin-Hebei (BTH) region is most serious during wintertime when the precipitation over BTH region is very small, thus we think the case we chose is more typical for the analysis of heavy haze over BTH region during wintertime. Moreover, since the main purpose of this paper is to introduce the QDA method, its applications to the heavy haze during summertime would be done in future work.

**(3)  As evaluated using NOR and SOR, I like the idea to consider the formation process from the viewpoint of each specie. The result of QDA is now discussed for PM2.5; however, each specie have been evolved as different E and C terms. I would like to strongly recommend to show the same kind of analysis of Figs. 7-9 for each specie. This analysis will offer the insight into C roles on chemical formation during haze episode.**

**Reply:** Thanks for this nice suggestion. Following the suggestion of reviewer, we analyze the QDA results for the secondary inorganic aerosols (SIAs), including nitrate, sulfate and ammonium, as well as their precursors, including NO$_x$, SO$_2$, and NH$_3$, to provide more insight into the C roles on the chemical formation during haze episode.

Figure R4 shows the QDA results for SIAs as well as their precursors during the different stages of episode. Note that since we parameterize 2.5% of sulfur emission as sulfate coatings on primary particles to consider the particle formation on sub-grid scale, thus there were small pure contribution of emission to the sulfate concentrations. As we can clearly see from Fig.R4, the chemical production of nitrate, sulfate and ammonium agreed well with the chemical depletion of their precursors, suggesting the good capability of the QDA method to represent the chemical processes in the model. For example, during the first stage, the values of C for NO$_x$, SO$_2$, and NH$_3$ were all negative where the C values for nitrate, sulfate and ammonium were positive, reflecting the conversion of reactive gases to the particulate matter. Consistent with the QDA results for PM$_{2.5}$ concentration, the QDA results for SIAs and their precursors shows that the chemistry provided an increasingly important role in the elevation of PM$_{2.5}$ concentrations. From stage 1 to stage 2, the values of C for NO$_x$, SO$_2$ changed from -0.18 to -0.27 $\mu g \cdot m^{-3} \cdot h^{-1}$ and from -0.01 to -0.02 $\mu g \cdot m^{-3} \cdot h^{-1}$ respectively. Correspondingly, the values of C for nitrate and sulfate increased from 0.21 to 0.26 $\mu g \cdot m^{-3} \cdot h^{-1}$ and from 0.02 to 0.03 $\mu g \cdot m^{-3} \cdot h^{-1}$, respectively. Consistent with the NOR and SOR analysis, the chemistry processes yield the largest contribution during stage 3 where the values of C for NO$_x$ and SO$_2$ up to -0.45 and -0.06 $\mu g \cdot m^{-3} \cdot h^{-1}$, respectively, which is 66.7% and almost twice higher than that during stage 2. Correspondingly, the C value for sulfate increased from 0.03 to 0.08 $\mu g \cdot m^{-3} \cdot h^{-1}$, almost twice higher than that during stage 2. However, the C value for nitrate was found to decrease in stage 3, which was only 0.07 $\mu g \cdot m^{-3} \cdot h^{-1}$, so did the C value for ammonium. In addition, the values of CE for nitrate and ammonium were much larger during stage 3 than those during stage 1 and stage 2, which were up to 0.46 and 0.15 $\mu g \cdot m^{-3} \cdot h^{-1}$, respectively. meanwhile, more NH$_3$ was also consumed by interaction between chemical and emission during stage 3 with CE value up to -0.15 $\mu g \cdot m^{-3} \cdot h^{-1}$. This is due to that the NH$_3$ is in poor condition during stage 3. Thus, although more NO$_x$ was oxidized to HNO$_3$ during stage 3 but most

of the newly formed $HNO_3$ were presented as gas phase due to the limited $NH_3$, leading to small C value for nitrate but large C value for $NO_x$. In addition, when there were emission processes during the simulation, the newly emitted NH3 would quickly react with the $HNO_3$ and form nitrate and ammonium. That's why the values of CE for nitrate and ammonium were much larger during stage 3. In the contrary, stage 1 and stage 2 were the NH3-rich condition so that the newly formed $HNO_3$ and $H_2SO_4$ can react with the existing $NH_3$ to form nitrate and sulfate without additional emission of $NH_3$. Therefore, it has a good consistence between the C values of precursors and those of SIAs during stage 1 and stage 2, and has small values of CE. These results suggest that the QDA method is capable of reflecting different chemical environment during different stages of episode, and emphasized that different emission control strategy should be taken during different stages of episode. For example, strict emission control should be performed for $NO_x$ and $SO_2$ emission during stage 1 and stage 2, while during stage 3 when the PM2.5 concentration was highest the control of $NH_3$ emission would be more efficient. This is in line with the results by Xu et al. (2019) who suggested that reducing $NH_3$ emission would be highly effective in reducing nitrate during severe winter haze events. Therefore, the QDA method can provide the policy maker with valuable insights into the development of efficient emission control strategy during different stages.

[Figure]

**Figure R4. The QDA results for $NO_x$ (a-d), $SO_2$ (e-h), $NH_3$ (i-l), nitrate (m-p), sulfate (q-t) and ammonium (u-x)**

**during different stages of the episode. Note that we used different scales for the contributions of M and E and those of other factors.**

**Comment 3:** The application of QDA. As found in the abstract, this QDA method could help modelers to understand each process and find these uncertainties. I have briefly checked the source code of QDA distributed in ZENODO, but I felt that the fortran90 codes seems to be incorporated into the NAQPMS source codes. How can we apply this source code into other models? If the authors claimed that "QDA is a universal tool", the explanation for how to use this QDA method in other models codes should be kindly introduced within this distribution.

**Reply:** We feel sorry for this confusion. "the QDA is a universal tool" is means that the algorithm of QDA can be applied in other models. In this paper, we only used the QDA method in the NAQPMS and developed the NAQPMS with QDA. To prevent ambiguity, we have revised this sentence as "To illustrate the use of QDA method, we developed a version of NAQPMS with QDA method. Also, the QDA method can be combined with different models following the algorithm of QDA".

**Minor comments:**

**Comment 4:** P2, L42: CMAQ have to be introduced after the definition of CTM (P2, L56). The organization of introduction for second and third paragraphs should be reconsidered.

**Reply:** Thanks for this suggestion. We will reconstruct the organization of the introduction for second and third paragraphs.

**Comment 5:** P2, L41-46: Under this context, IRR should be also carefully introduced. The IRR can be used to define the role of reaction rate, and this will relate C term in this study.

**Reply:** Thanks for this suggestion. we have revised this sentence as "The integrated process rate (IPR) considered in the Community Multiscale Air Quality (CMAQ) model can be used to define the role of reaction rate, and thus provides implications for the contributions of chemical processes to the formation of air pollution".

**Comment 6:** P4, L92-L99 and Eq. 2: How can we treat the second- and third-order partial differential of x1. x2, and x3? Does this represent the nonlinear term of M, E, and C? What stands for them?

**Reply:** Thanks for this comment. As we responded in Comment 1, the second- and higher-order partial differential of x1, x2, and x3, together with their first-order partial differential are defined as the pure contribution of emission, meteorology and chemistry.

**Comment 7:** P4, L104: For example, higher temperature will relate activated plant, and change the biogenic emissions intensity. Why E to M is unidirectional?

**Reply:** Thanks for this comment. We have revised this sentence as "Note that although the effects of emissions and meteorology is bi-directional, for example the higher temperatures would increase evaporative emissions from gasoline vehicles, the effect of emissions on meteorology is unidirectional in our application since we did not have an online

emission model to represent the interactions between emissions and meteorology, which would be a limitation of our work."

**Comment 8:** P4, L112-113: As commented in major point 1, how did conduct accompanying simulations at each time step? The detailed description of each scenario should be explained.

**Reply:** Thanks for this comment. please refer to our responses to Comment 1 for detailed description of how to conduct the accompanying simulation.

**Comment 9:** P4, L113-115: However, even though each accompanying simulation conducted at each time step, the result is merely derived from the difference (subtraction) from baseline simulation. What was the advantages to embed these accompanying simulations? How about the computational burden? It was not clearly stated here. Therefore, I cannot follow the importance of QDA method as novel way.

**Reply:** Thanks for this comment. As well illustrated in our responses to Comment 1, embedding the accompanying simulations is necessary for the QDA method, which facilitates the updates of the initial condition of accompanying simulations and enables the calculation of contributions of different factors as well as their interactions to the variation of model results. Please refer to our responses to Comment 1 for detailed description on the accompanying simulation. Since there were six accompanying simulations, thus the computational burden for QDA would increase by six times.

**Comment 10**: P4, L119-120: In case of sulfate, this will be also produced in aqueous-phase oxidation pathways. How this process was incorporated?

**Reply:** We feel Sorry for this confusion. The aqueous-phase oxidation pathways of sulfate have been incorporated in gas-phase chemistry. For clarity, we have revised this sentence as "The above QDA method can also be combined with the IPR method to resolve more detailed information, such as the contributions of advection, diffusion, dry and wet deposition processes in M or the contributions of the gas- and aqueous-phase chemistry, thermodynamic equilibrium processes and reactions involving secondary organic aerosols (SOAs) in C".

**Comment 11:** P5, Section 2.2: The core mechanisms configured NAQPMS seems to be outdated over 20 years as stated in this section. Despite the recent progress of modeling components, I cannot follow "… has been widely used in scientific research and air quality prediction practice (Wang et al., 2014) due to its good performance in simulating the emission, meteorological and chemical processes in the atmosphere.". Detailed introductions of research examples are required, because the modeling performance itself will be important to discuss this manuscript.

**Reply:** Thanks for this comment. we have added more detailed introductions of research examples in the revised manuscript to illustrate the performance of the model.

**Comment 12:** P5, L132: Typo in "ISORRPIA".

**Reply:** Done.

**Comment 13:** P5, L142: What was this year? It should be defined first here.

**Reply:** We feel sorry for this confusion. This year is 2014.

**Comment 14:** P5, L143-144: What was the height of lowermost layer? It should be explicitly stated to consider the modeling performance at surface level.

**Reply:** Thanks for this comment. The height of lowermost layer is about 50m, which has been clarified it in the revised manuscript.

**Comment 15:** P5, L145: Was the MEIC also targeted to the analyzed year?

**Reply:** We feel sorry for this confusion. The base year of MEIC emission inventory that we used is 2014, and we have clarified it in the revised manuscript.

**Comment 16:** P5, L147: What kind of biomass burning emissions was used? If not used, why?

**Reply:** We feel sorry for our carelessness. The biomass burning emissions is obtained from GFED4, which has been clarified in the revised manuscript.

**Comment 17:** P5, L149: Confirm the version of MOZART 2.4 or 2.5?

**Reply:** We feel sorry for this confusion. The version of MOZART is 2.5.

**Comment 18:** P5, L151: Again, WRF version 3.7 seems to be also outdated. What is the exact reason to use this version to generate meteorological field despite the authors' claim of the importance of meteorology.

**Reply:** The use of WRF version 3.7 is because we think it is a stable version of WRF which has been widely used in previous studies.

**Comment 19:** P5, L150-152: Does NAQPMS model online-coupled to WRF meteorological field? It was not clarified here.

**Reply:** We feel sorry for this confusion. We have clarified that the NAQPMS does not online-coupled to the WRF simulation in the revised manuscript.

**Comment 20:** P6, L171: Need the definition of LST. What is the difference from GMT?

**Reply:** We feel sorry for this confusion. The LST means the local standard time. We have clarified it in the revised manuscript.

**Comment 21:** P7. L193-194: Over China, recommendations of modeling standards have been updated (https://acp.copernicus.org/articles/21/2725/2021/), and it is better to use this criteria because this study targeted BTH region.

**Reply:** Thanks for this suggestion. Following the suggestion of reviewer, the goals and criteria proposed by Huang et al. (2021) are used to illuminate the robustness and reliability of our mode results in the revised manuscript. The results suggest that the simulated total $PM_{2.5}$ concentrations all satisfied the normalized mean bias (NMB), normalized mean error (NBE), correlation coefficients (R), and index of agreement (IOA) performance standards (NMB<20%, NBE<45%, R<0.6 and IOA > 0.7).

**Comment 22:** Figure 5 and 9: Without the explicit information of vertical layer height, this presentation is weak. Please clarify these information on Section 2.2 or 2.3.

**Reply:** We feel sorry for this confusion. We have added the explicit information of the vertical height in the revised

manuscript.

**Comment 23:** Figure 8: The contribution of M and E terms are larger compared to other terms. I would like to recommend to use different scale for them, especially for (e)-(h). Again as I have commented as major comments of 1 and 2, this result impressed me that QDA was just the usage of IPR method. Please clarify this point in introduction and methodology.

**Reply:** Thanks for this suggestion. We have revised the figure 8 by using different scale for the M, E and other factors.

**Comment 24:** Figure 10: Was the vertical axis used log-scale? It seems to be used unusually scaled axis.

**Reply:** We feel sorry for this confusion. The vertical axis of figure 10 used sigma-p vertical axis labeled by the height of different layers.

**References:**

Huang, L., Zhu, Y., Zhai, H., Xue, S., Zhu, T., Shao, Y., Liu, Z., Emery, C., Yarwood, G., Wang, Y., Fu, J., Zhang, K., and Li, L.: Recommendations on benchmarks for numerical air quality model applications in China – Part 1: PM2.5 and chemical species, Atmos. Chem. Phys., 21, 2725-2743, 10.5194/acp-21-2725-2021, 2021.

Li, M., Zhang, Q., Kurokawa, J. I., Woo, J. H., He, K., Lu, Z., Ohara, T., Song, Y., Streets, D. G., Carmichael, G. R., Cheng, Y., Hong, C., Huo, H., Jiang, X., Kang, S., Liu, F., Su, H., and Zheng, B.: MIX: a mosaic Asian anthropogenic emission inventory under the international collaboration framework of the MICS-Asia and HTAP, Atmos. Chem. Phys., 17, 935-963, 10.5194/acp-17-935-2017, 2017.

Stein, U. and Alpert, P.: FACTOR SEPARATION IN NUMERICAL SIMULATIONS, Journal of the Atmospheric Sciences, 50, 2107-2115, 10.1175/1520-0469(1993)050<2107:Fsins>2.0.Co;2, 1993.

Xu, Z., Liu, M., Zhang, M., Song, Y., Wang, S., Zhang, L., Xu, T., Wang, T., Yan, C., Zhou, T., Sun, Y., Pan, Y., Hu, M., Zheng, M., and Zhu, T.: High efficiency of livestock ammonia emission controls in alleviating particulate nitrate during a severe winter haze episode in northern China, Atmos. Chem. Phys., 19, 5605-5613, 10.5194/acp-19-5605-2019, 2019.